# ACTOR-CRITIC IS IMPLICITLY BIASED TOWARDS HIGH ENTROPY OPTIMAL POLICIES

**Yuzheng Hu,  Ziwei Ji,  Matus Telgarsky**
University of Illinois, Urbana-Champaign
`<{yh46,ziweiji2,mjt}@illinois.edu>`

## ABSTRACT

We show that the simplest actor-critic method — a linear softmax policy updated with TD through interaction with a linear MDP, but featuring no explicit regularization or exploration — does not merely find an optimal policy, but moreover prefers high entropy optimal policies. To demonstrate the strength of this bias, the algorithm not only has no regularization, no projections, and no exploration like $\epsilon$-greedy, but is moreover trained on a single trajectory with no resets. The key consequence of the high entropy bias is that uniform mixing assumptions on the MDP, which exist in some form in all prior work, can be dropped: the implicit regularization of the high entropy bias is enough to ensure that all chains mix and an optimal policy is reached with high probability. As auxiliary contributions, this work decouples concerns between the actor and critic by writing the actor update as an explicit mirror descent, provides tools to uniformly bound mixing times within KL balls of policy space, and provides a projection-free TD analysis with its own implicit bias which can be run from an unmixed starting distribution.

## 1 OVERVIEW

Reinforcement learning methods navigate an environment and seek to maximize their reward (Sutton & Barto, 2018). A key tension is the tradeoff between *exploration* and *exploitation*: does a learner (also called an *agent* or *policy*) *explore* for new high-reward states, or does it *exploit* the best states it has already found? This is a sensitive part of RL algorithm design, as it is easy for methods to become blind to parts of the state space; to combat this, many methods have an explicit exploration component, for instance the *$\epsilon$-greedy method*, which forces exploration in all states with probability $\epsilon$ (Sutton & Barto, 2018; Tokic, 2010). Similarly, many methods must use projections and regularization to smooth their estimates (Williams & Peng, 1991; Mnih et al., 2016; Cen et al., 2020).

This work considers actor-critic methods, where a policy (or *actor*) is updated via the suggestions of a *critic*. In this setting, prior work invokes a combination of explicit regularization and exploration to avoid getting stuck, and makes various fast mixing assumptions to help accurate exploration. For example, recent work with a single trajectory in the tabular case used both an explicit $\epsilon$-greedy component and uniform mixing assumptions (Khodadadian et al., 2021), neural actor-critic methods use a combination of projections and regularization together with various assumptions on mixing and on the path followed through policy space (Cai et al., 2019; Wang et al., 2019), and even direct analyses of the TD subroutine in our linear MDP setting make use of both projection steps and an assumption of starting from the stationary distribution (Bhandari et al., 2018).

**Contribution.** This work shows that a simple linear actor-critic (cf. Algorithm 1) in a linear MDP (cf. Assumption 1.3) with a finite but non-tabular state space (cf. Assumption 1.1) finds an $\epsilon$-optimal policy in $\text{poly}(1/\epsilon)$ samples, without any explicit exploration or projections in the algorithm and without any uniform mixing assumptions on the policy space (cf. Theorem 1.4). The algorithm and analysis avoid both via an *implicit bias* towards high entropy policies: the actor-critic policy path never leaves a Kullback-Leibler (KL) divergence ball of the maximum entropy optimal policy, and this firstly ensures implicit exploration, and secondly ensures fast mixing. In more detail:

1. **Actor analysis via mirror descent.** We write the actor update as an explicit mirror descent. While on the surface this does not change the method (e.g., in the tabular case, the method is identical to natural policy gradient (Agarwal et al., 2021b)), it gives a clean optimization guarantee which carries a KL-based implicit bias consequence for free, and decouples concerns between the actor and critic.

2. **Critic analysis via projection-free sampling tools within KL balls.** The preceding mirror descent component guarantees that we stay within a small KL ball, *if* the statistical error of the critic is controlled. Concordantly, our sampling tools guarantee this statistical error is small, *if* we stay within a small KL ball. Concretely, we provide useful lemmas that every policy in a KL ball around the high entropy policy has uniformly upper bounded mixing times, and separately give a projection-free (implicitly regularized!) analysis of the standard *temporal-difference (TD)* update from any starting state (Sutton, 1988), whereas the closest TD analysis in the literature uses projections and requires the sampling process to be started from the stationary distribution (Bhandari et al., 2018). The mixing assumptions here contrast in general with prior work, which either makes explicit use of stationary distributions (Cai et al., 2019; Wang et al., 2019; Bhandari et al., 2018), or makes uniform mixing assumptions on all policies (Xu et al., 2020; Khodadadian et al., 2021).

In addition to the preceding major contributions, the paper comes with many technical lemmas (e.g., mixing time lemmas) which we hope are useful in other work.

## 1.1 SETTING AND MAIN RESULTS

We will now give the setting, main result, and algorithm in full. Further details on MDPs can be found in Section 1.3, but the actor-critic method appears in Algorithm 1. To start, the environment and policies are as follows.

**Assumption 1.1.** The *Markov Decision Process (MDP)* has states $s \in \mathbb{R}^d$ and finitely many actions $a \in \mathcal{A} := \{e_1, \ldots, e_k\}$, and finite rewards $r \in [0, 1]$. States are observed in some feature encoding $s \in \mathbb{R}^d$, but the state space $\mathcal{S} \subseteq \{s \in \mathbb{R}^d : 1/2 \le \|s\| \le 1\}$ is assumed finite: $|\mathcal{S}| < \infty$.

Policies are *linear softmax policies*: a policy is parameterized by a weight matrix $W \in \mathbb{R}^{d \times k}$, and given a state $s \in \mathbb{R}^d$, uses a per-state softmax to sample a new action $a$:

$$a \sim \phi(s^\mathsf{T} W \cdot), \qquad \text{where } \phi(s^\mathsf{T} W a) = \frac{\exp(s^\mathsf{T} W a)}{\sum_{b \in \mathcal{A}} \exp(s^\mathsf{T} W b)}. \tag{1.1}$$

Let $\mathcal{A}_s$ denote the set of optimal actions for a given state $s$. It is assumed that $\mathcal{A}_s$ is nonempty for every $s \in \mathcal{S}$, and that there exists at least one optimal policy which is *irreducible* (Levin et al., 2006, Chapter 1). $\diamondsuit$

The choice of linear policies simplifies presentation and analysis, but the tools here should be applicable to other settings. This choice also allows direct comparison to the widely-studied implicit bias

---

**Algorithm 1** Single-trajectory linear actor-critic.

---

**Inputs:** actor iterations $t$ and step size $\theta$; critic iterations $N$ and step size $\eta$.
**Initialize:** actor weights $W_0 = 0 \in \mathbb{R}^{d \times k}$, pre-softmax mapping $p_0(s, a) := s^\mathsf{T} W_0 a$, policy $\pi_0 := \phi(p_0)$ (cf. eq. (1.1)); sample initial state/action/reward triple $(s_{0,0}, a_{0,0}, r_{0,0})$.
**for** $i = 0, 1, 2, \ldots, t-1$ **do**
  **Critic update:** use $\pi_i$ to interact with the MDP, obtaining state/action/reward triples $(s_{i,j}, a_{i,j}, r_{i,j})_{j \le N}$ by continuing the existing trajectory, and form *TD estimates*

  $$U_{i,j+1} := U_{i,j} - \eta s_{i,j} \left( s_{i,j}^\mathsf{T} U_{i,j} a_{i,j} - \gamma s_{i,j+1}^\mathsf{T} U_{i,j} a_{i,j+1} - r_{i,j} \right) a_{i,j}^\mathsf{T},$$

  with initial condition $U_{i,0} = 0$. Set $\widehat{U}_i := \frac{1}{N} \sum_{j < N} U_{i,j}$ and $\widehat{\mathcal{Q}}_i(s, a) := s^\mathsf{T} \widehat{U}_i a$, and also $(s_{i+1,0}, a_{i+1,0}, r_{i+1,0}) := (s_{i,N}, a_{i,N}, r_{i,N})$ to continue the existing trajectory in next iteration.
  **Actor update:** set $W_{i+1} := W_i + \theta \widehat{U}_i$ and $p_{i+1} := p_i + \theta \widehat{\mathcal{Q}}_i$ and $\pi_{i+1} := \phi(p_{i+1})$.
**end for**

---

of gradient descent in linear classification settings (Soudry et al., 2017; Ji & Telgarsky, 2018), as will be discussed further in Section 1.2. The choice of finite state space is to remove measure-theoretic concerns and to allow a simple characterization of the maximum entropy optimal policy.

**Lemma 1.2** (simplification of Lemma A.1). *Under Assumption 1.1, there exists a unique maximum entropy policy $\overline{\pi}$, which satisfies $\overline{\pi}(s, \cdot) = \mathrm{Uniform}(\mathcal{A}_s)$ for every state $s$, and moreover has a stationary distribution $\mathfrak{s}_{\overline{\pi}}$.*

To round out this introductory presentation of the actor, the last component is the update: $p_{i+1} := p_i + \widehat{\mathcal{Q}}_i$ and $\pi_{i+1} := \phi(p_{i+1})$, where $\widehat{\mathcal{Q}}_i$ is the *TD estimate of the Q function*, to be discussed shortly. This update is explicitly a *mirror descent* or *dual averaging* update of the policy, where we use a *mirror mapping* $\phi$ to obtain the policy $\pi_{i+1}$ from pre-softmax values $p_{i+1}$. As mentioned before, this update appears in prior work in the tabular setting with natural policy gradient and actor-critic (Agarwal et al., 2021b; Khodadadian et al., 2021), and will be related to other methods in Section 1.2. We will further motivate this update in Section 2.

The final assumption and description of the critic are as follows. As will be discussed in Section 2, the policy becomes optimal if $\widehat{\mathcal{Q}}_i$ is an accurate estimate of the true $Q$ function. We employ a standard TD update with no projections or constraints. To guarantee that this linear model of $\mathcal{Q}_i$ is accurate, we make a standard *linear MDP assumption* (Bradtke & Barto, 1996; Melo & Ribeiro, 2007; Jin et al., 2020).

**Assumption 1.3.** In words, the *linear MDP assumption* is that the MDP rewards and transitions are modeled by linear functions. In more detail, for convenience first fix a canonical vector form for state/action pairs $(s, a) \in \mathbb{R}^{d \times k}$: let $x_{sa} \in \mathbb{R}^{dk}$ denote the vector obtained via unrolling the matrix $sa^\intercal$ row-wise (whereby vector inner products with $x_{sa}$ match matrix inner products with $sa^\intercal$). The linear MDP assumption is then that there exists a fixed vector $y \in \mathbb{R}^{dk}$ and a fixed matrix $M \in \mathbb{R}^{d \times dk}$ so that for any state/action pair $x_{sa}$ and any subsequent state $s' \in \mathbb{R}^d$,

$$\mathbb{E}[r \,|\, (s, a)] = x_{sa}^\intercal y, \qquad \text{and} \qquad \mathbb{E}[s' \,|\, (s, a)] = M x_{sa}.$$

Lastly, suppose $1/2 \leq \|s\| \leq 1$ for all $s \in \mathcal{S}$. $\diamondsuit$

Though a strong assumption, it is not only common, but note also that since TD must continually interact with the MDP, then it would have little hope of accuracy if it can not model short-term MDP dynamics. Indeed, as is shown in Lemma C.3 (but appears in various forms throughout the literature), Assumption 1.3 implies that the fixed point of the TD update is the true $Q$ function. This assumption and the closely related *compatible linear function approximation* assumption will be discussed in Section 1.2.

We now state our main result, which bounds not just the *value function* $\mathcal{V}$ (cf. Section 1.3) but also the KL divergence $K_{\mathfrak{v}_{\overline{\pi}}^s}(\overline{\pi}, \pi_i) = \mathbb{E}_{s' \sim \mathfrak{v}_{\overline{\pi}}^s} \sum_a \overline{\pi}(s', a) \ln \frac{\overline{\pi}(s', a)}{\pi_i(s', a)}$, where $\mathfrak{v}_{\overline{\pi}}^s$ is the *visitation distribution* of the maximum entropy optimal policy $\overline{\pi}$ when run from state $s$ (cf. Section 1.3).

**Theorem 1.4.** *Suppose Assumptions 1.1 and 1.3 (which imply the (unique) maximum entropy optimal policy $\overline{\pi}$ is well-defined and has a stationary distribution). Given iteration budget $t$, choose*

$$\theta = \Theta\left(\frac{1}{t^{13/16} \ln(t)^{1/4}}\right), \qquad N = \Theta\left(t^2 \ln t\right), \qquad \eta = \Theta\left(\frac{1}{\sqrt{N \ln N}}\right),$$

*where the constants hidden inside each $\Theta(\cdot)$ depend only on $\overline{\pi}$ and the MDP, but not on $t$. With these parameters in place, invoke Algorithm 1, and let $(\pi_i)_{i<t}$ be the resulting sequence of policies. Then with probability at least $1 - 1/t^{1/8}$, simultaneously for every state $s \in \mathbb{R}^d$ and every $i \leq t$,*

$$K_{\mathfrak{v}_{\overline{\pi}}^s}(\overline{\pi}, \pi_i) + \theta(1 - \gamma) \sum_{j<i} \left(\mathcal{V}_{\overline{\pi}}(s) - \mathcal{V}_j(s)\right) \leq \ln k + \frac{1}{(1 - \gamma)^2}.$$

Before outlining the proof structure and organization of the rest of the paper, a few comments on the interpretation of Theorem 1.4 are as follows.

**Remark 1.5** (Discussion of Theorem 1.4).

1. **Implicit bias.** Since $\overline{\pi}$ is optimal, the second term can be deleted, and the bound implies

$$\max_{\substack{i \leq t \\ s \in \mathcal{S}}} K_{\mathfrak{v}_{\overline{\pi}}^s}(\overline{\pi}, \pi_i) \leq \ln k + \frac{1}{(1 - \gamma)^2};$$

since this holds for all $i \leq t$, it controls the optimization path. This term is a direct consequence of our mirror descent setup, and is used to control the TD errors at every iteration. This implicit bias of the policy path stands therefore in stark contrast to the worst-case KL divergence between *arbitrary* softmax policies and $\overline{\pi}$, which is infinite: e.g., a sequence of policies $(\pi_i)_{i \geq 1}$ which place vanishing probability on a pair $(s, a)$ which in turn receives positive probability under $\overline{\pi}$ will have $K_{\mathfrak{v}_{\overline{\pi}}^s}(\overline{\pi}, \pi_i) \to \infty$.

2. **Mixing time constants.** The critic iterations $N$ and step size $\eta$ hide mixing time constants; these mixing time constants depend only on the KL bound $\ln k + 1/(1-\gamma)^2$, and in particular there is no hidden growth in these terms with $t$. That is to say, mixing times are uniformly controlled over a fixed KL ball that does not depend on $t$; prior work by contrast makes strong mixing assumptions (Wang et al., 2019; Xu et al., 2020; Khodadadian et al., 2021).

3. **High probability guarantee.** Though prior work focuses on bounds in expectation, we chose a high probability guarantee to emphasize that the bound does not blow up, despite an arguably more strenuous setting.

4. **Single trajectory.** A single trajectory through the MDP is used to remove the option of the algorithm escaping from poor choices with resets; only the implicit bias can save it.

5. **Rate.** To reach a policy which whose value function is $\epsilon$-close to optimal, a trajectory length (number of samples) of $1/\epsilon^{16}$ is sufficient, ignoring log factors (to obtain this from Theorem 1.4, it suffices to divide both sides of the bound by $t\theta(1-\gamma)$, set $t = 1/\epsilon^{16/3}$, and note the trajectory length is $tN$). This is slower than the $1/\epsilon^6$ given in the only other single-trajectory analysis in the literature Khodadadian et al. (2021), but by contrast that work makes uniform mixing assumptions (cf. Khodadadian et al. (2021, Lemma C.1)), requires the tabular setting, and uses $\epsilon$-greedy for explicit exploration in each iteration. $\diamondsuit$

The proof of Theorem 1.4 and organization of the paper are as follows. After overviews of related work and notation in Sections 1.2 and 1.3, the first proof component, discussed in Section 2, is the outer loop of Algorithm 1, namely the update to the policy $\pi_i$. As discussed before, this part of the analysis writes the policy update as a mirror descent, and conveniently decouples the suboptimality error into an *actor error*, which is handled by standard mirror descent tools and provides the implicit bias towards high entropy policies, and a *critic error*, namely the error of the estimated $Q$ function $\widehat{\mathcal{Q}}_i$. The second component, discussed in Section 3, is therefore the TD analysis establishing that the estimate $\widehat{\mathcal{Q}}_i$ is accurate, which not only requires an abstract TD guarantee (which, as mentioned, is projection-free and run from an arbitrary starting state, unlike prior work), but also requires tools to explicitly bound mixing times, rather than assuming mixing times are bounded.

This culminates in the proof of Theorem 1.4, which is sketched at the end of Section 3, and uses an induction combining the guarantees from the preceding two proof components at all times: it is established inductively that the next policy $\pi_{i+1}$ has high entropy and low suboptimality because the previous $Q$ function estimates $(\widehat{\mathcal{Q}}_j)_{j \leq i}$ were accurate, and simultaneously that the next estimate $\widehat{\mathcal{Q}}_{i+1}$ is accurate because $\pi_{i+1}$ has high entropy, which guarantees fast mixing. Section 4 concludes with some discussion and open problems, and the appendices contain the full proofs.

## 1.2 Further related work

For the standard background in reinforcement learning, see the book by Sutton & Barto (2018). Standard concepts and notation choices are presented below in Section 1.3.

**Standard algorithms: PG/NPG and AC/NAC.** The *[natural] policy gradient ([N]PG)* and *natural actor-critic ([N]AC)* are widely used in practice, and summarized briefly as follows. Policy gradient methods update the actor parameters $W$ with gradient ascent on the value function $\mathcal{V}_\pi(\mu)$ for some state distribution $\mu$ (Williams, 1992; Sutton et al., 2000; Bagnell & Schneider, 2003; Liu et al., 2020; Fazel et al., 2018), whereas natural policy gradient multiplies $\nabla_W \mathcal{V}_\pi(\mu)$ by an *inverse Fisher matrix* with the goal of improved convergence via a more relevant geometry (Kakade, 2001; Agarwal et al., 2021b). What policy gradient leaves open is how to estimate $\nabla_W \mathcal{V}_\pi(\mu)$; actor-critic methods go one step further and suggest updating the actor with policy gradient as above, but noting that $\nabla_W \mathcal{V}_\pi(\mu)$ can be written as a function of $\mathcal{Q}_\pi$, or rather an estimate thereof, and making this

estimation the job of a separate subroutine, called the *critic* Konda & Tsitsiklis (2000). (*Natural* actor-critic uses *natural* policy gradient in the actor update (Peters & Schaal, 2008).) Actor-critic methods are perhaps the most widely-used instances of policy gradient, and come in many forms; the use of TD for the critic step is common (Williams, 1992; Sutton et al., 2000; Bagnell & Schneider, 2003; Liu et al., 2020; Fazel et al., 2018).

**Linear MDPs and compatible function approximation.** The linear MDP assumption (cf. Assumption 1.3) is used here to ensure that the TD step accurately estimates the true $Q$ function, and is a somewhat common assumption in the literature, even when TD is not used (Bradtke & Barto, 1996; Melo & Ribeiro, 2007; Jin et al., 2020). As an example, the *tabular* setting satisfies Assumption 1.3, simply by encoding states as distinct standard basis vectors, namely $\mathcal{S} := \{e_1, \ldots, e_{|\mathcal{S}|}\}$ (Jin et al., 2020, Example 2.1); moreover, in this tabular setting, the actor update of Algorithm 1 agrees with NPG (Agarwal et al., 2021b). Interestingly, another common assumption, *compatible linear function approximation*, also guarantees our analysis goes through and that Algorithm 1 agrees with NPG, while being non-tabular in general. In detail, the compatible function approximation setting firstly requires that the actor update agrees with $\widehat{\mathcal{Q}}_i$ in a certain sense (which holds in our setting by construction), and secondly that there exists a choice of critic parameters $U$ so that the exact $Q$ function $\mathcal{Q}_\pi$ can be represented with these parameters (Silver, 2015). If this assumption holds *for every policy $\pi_i$ (and corresponding $\mathcal{Q}_i$) encountered in the algorithm*, then the policy update of Algorithm 1 agrees with NPG and NAC (this is a standard fact; see for instance Silver (2015), or Agarwal et al. (2021a, Lemma 13.1)). Additionally, the proofs here also go through under this arguably weaker assumption: Assumption 1.3 is only used to ensure that TD finds not just a fixed point but the true $Q$ function (cf. Lemma C.4), which is also guaranteed by the compatibility assumption. However, since this is an assumption on the trajectory and thus harder to interpret, we prefer Assumption 1.3 which explicitly holds for all possible policies.

**Regularization and constraints.** It is standard with neural policies to explicitly maintain a constraint on the network weights (Wang et al., 2019; Cai et al., 2019). Relatedly, many works both in theory and practice use explicit entropy regularization to prevent small probabilities (Williams & Peng, 1991; Mnih et al., 2016; Abdolmaleki et al., 2018), and which can seem to yield convergence rate improvements (Cen et al., 2020).

**NPG and mirror descent.** (For background on mirror descent, see Section 2 and appendix B.) The original and recent analyses of NPG had a mirror descent flavor, though mirror descent and its analysis were not explicitly invoked (Kakade, 2001; Agarwal et al., 2021b). Further connections to mirror descent have appeared many times (Geist et al., 2019; Shani et al., 2020), though with a focus on the design of new algorithms, and not for any implicit regularization effect or proof. Mirror descent is used heavily throughout the online learning literature (Shalev-Shwartz, 2011), and in work handling adversarial MDP settings (Zimin & Neu, 2013).

**Temporal-difference update (TD).** As discussed before, the TD update, originally presented by (Sutton, 1988), is standard in the actor-critic literature (Cai et al., 2019; Wang et al., 2019), and also appears in many other works cited in this section. As was mentioned, prior work requires various projections and initial state assumptions (Bhandari et al., 2018), or positive eigenvalue assumptions (Zou et al., 2019; Srikant & Ying, 2019; Bhandari et al., 2018).

**Implicit regularization in supervised learning.** A pervasive topic in supervised learning is the *implicit regularization* effect of common descent methods; concretely, standard descent methods prefer low or even minimum norm solutions, which can be converted into generalization bounds. The present work makes use of a *weak* implicit bias, which only prefers smaller norms and does not necessarily lead to minimal norms; arguably this idea was used in the classical perceptron method (Novikoff, 1962), but was then shown in linear and shallow network cases of SGD applied to logistic regression (Ji & Telgarsky, 2018; 2019), which was then generalized to other losses (Shamir, 2020), and also applied to other settings (Chen et al., 2019). The more well-known *strong* implicit bias, namely the convergence to minimum norm solutions, has been observed with exponentially-tailed losses together with coordinate descent with linear predictors (Zhang & Yu, 2005; Telgarsky, 2013), gradient descent with linear predictors (Soudry et al., 2017; Ji & Telgarsky, 2018), and deep learning in various settings (Lyu & Li, 2019; Chizat & Bach, 2020), just to name a few.

## 1.3 NOTATION

This brief notation section summarizes various concepts and notation used throughout; modulo a few inventions, the presentation mostly matches standard ones in RL (Sutton & Barto, 2018) and policy gradient (Agarwal et al., 2021b). A policy $\pi : \mathbb{R}^d \times \mathbb{R}^k \to \mathbb{R}$ maps state-action pairs to reals, and $\pi(s, \cdot)$ will always be a probability distribution. Given a state, the agent samples an action from $a \sim \pi(s, \cdot)$, the environment returns some random reward (which has a fixed distribution conditioned on the observed $(s, a)$ pair), and then uses a *transition kernel* to choose a new state given $(s, a)$.

Taking $\tau$ to denote a random trajectory followed by a policy $\pi$ interacting with the MDP from an arbitrary initial state distribution $\mu$, the *value* $\mathcal{V}$ and *Q functions* are respectively

$$\mathcal{V}_\pi(\mu) := \mathbb{E}_{\substack{s_0 \sim \mu \\ \tau = s_0, a_0, r_0, s_1, \cdots}} \sum_{t \geq 0} \gamma^t r_t,$$

$$\mathcal{Q}_\pi(s, a) := \mathbb{E}_{\substack{s_0 = s, a_0 = a \\ \tau = s_0, a_0, r_0, s_1, \cdots}} \sum_{t \geq 0} \gamma^t r_t = \mathbb{E}_{r_0 \sim (s,a)} \left( r_0 + \gamma \mathbb{E}_{s_1 \sim (s,a)} \mathcal{V}_\pi(s_1) \right),$$

where the simplified notation $\mathcal{V}_\pi(s) = \mathcal{V}_\pi(\delta_s)$ for Dirac distribution $\delta_s$ on state $s$ will often be used, as well as the shorthand $\mathcal{V}_i = \mathcal{V}_{\pi_i}$ and $\mathcal{Q}_i = \mathcal{Q}_{\pi_i}$. Additionally, let $\mathcal{A}_\pi(s, a) := \mathcal{Q}_\pi(s, a) - \mathcal{V}_\pi(s)$ denote the *advantage function*; note that the natural policy gradient update could interchangeably use $\mathcal{A}_i$ or $\mathcal{Q}_i$ since they only differ by an action-independent constant, namely $\mathcal{V}_\pi(s)$, which the softmax normalizes out. As in Assumption 1.1, the state space $\mathcal{S}$ is finite but a subset of $\mathbb{R}^d$, specifically $\mathcal{S} \subseteq \{s \in \mathbb{R}^d : 1/2 \leq \|s\| \leq 1\}$, and the action space $\mathcal{A}$ is just the $k$ standard basis vectors $\{e_1, \ldots, e_k\}$. The other MDP assumption, namely of a *linear MDP* (cf. Assumption 1.3), will be used whenever TD guarantees are needed. Lastly, the *discount factor* $\gamma \in (0, 1)$ has not been highlighted, but is standard in the RL literature, and will be treated as given and fixed throughout the present work.

A common tool in RL is the *performance difference lemma* (Kakade & Langford, 2002): letting $\mathfrak{v}_\pi^\mu$ denote the *visitation distribution* corresponding to policy $\pi$ starting from $\mu$, meaning

$$\mathfrak{v}_\pi^\mu := \frac{1}{1 - \gamma} \mathbb{E}_{s' \sim \mu} \sum_{t \geq 0} \gamma^t \Pr[s_t = s | s_0 = s'],$$

the performance difference lemma can be written as

$$\mathcal{V}_\pi(\mu) - \mathcal{V}_{\pi'}(\mu) = \frac{1}{1 - \gamma} \mathbb{E}_{s \sim \mathfrak{v}_{\pi'}^\mu} \sum_a \mathcal{Q}_\pi(s, a)(\pi(s, a) - \pi'(s, a)) =: \frac{1}{1 - \gamma} \left\langle \mathcal{Q}_\pi, \pi - \pi' \right\rangle_{\mathfrak{v}_{\pi'}^\mu},$$

(1.2)

where the final inner product notation will often be employed for convenience.

In a few places, we need the Markov chain *on states*, $P_\pi$, which is induced by a policy $\pi$: that is, the chain where given a state $s$, we sample $a \sim \pi(s, \cdot)$, and then transition to $s' \sim (s, a)$, where the latter sampling is via the MDP's transition kernel.

As mentioned above, $\mathfrak{s}_\pi$ will denote the stationary distribution of a policy $\pi$ whenever it exists. The only relevant assumption we make here, namely Assumption 1.1, is that the maximum entropy optimal policy $\overline{\pi}$ is aperiodic and irreducible, which implies it has a stationary distribution with positive mass on every state (Levin et al., 2006, Chapter 1). Via Lemma 3.1, it follows that all policies in a KL ball around $\overline{\pi}$ also have stationary distributions with positive mass on every state.

The max entropy optimal policy $\overline{\pi}$ is complemented by a (unique) optimal $Q$ function $\overline{\mathcal{Q}}$ and optimal advantage function $\overline{\mathcal{A}}$. The optimal $Q$ function $\overline{\mathcal{Q}}$ dominates all other $Q$ functions, meaning $\overline{\mathcal{Q}}(s, a) \geq \mathcal{Q}_\pi(s, a)$ for any policy $\pi$; for details and a proof, see Lemma A.1.

We use $\|\mu - \nu\|_{\text{TV}} = \sup_{U \subseteq \mathcal{S}} |\mu(U) - \nu(U)|$ to denote the *total variation distance*, which is pervasive in mixing time analyses (Levin et al., 2006).

## 2 MIRROR DESCENT TOOLS

To see how nicely mirror descent and its guarantees fit with the NPG/NAC setup, first recall our updates: $p_{i+1} := p_i + \widehat{\mathcal{Q}}_i$, and $\pi_{i+1} := \phi(p_{i+1})$ (e.g., matching NPG in the tabular case (Kakade,

2001; Agarwal et al., 2021b)). In the online learning literature (Shalev-Shwartz, 2011; Lattimore & Szepesvári, 2020), the basic mirror ascent (or *dual averaging*) guarantee is of the form

$$\sum_{i<t} \left\langle \widehat{\mathcal{Q}}_i, \pi - \pi_i \right\rangle = \mathcal{O}\left(\sqrt{t}\right),$$

where notably $\widehat{\mathcal{Q}}_i$ can be an arbitrary matrix. The most common results are stated when $\widehat{\mathcal{Q}}_i$ is the gradient of some convex function, but here instead we can use the performance difference lemma (cf. eq. (1.2)): recalling the inner product and visitation distribution notation from Section 1.3,

$$\left\langle \widehat{\mathcal{Q}}_i, \pi_i - \pi \right\rangle_{\mathfrak{v}_\pi^\mu} = \langle \mathcal{Q}_i, \pi_i - \pi \rangle_{\mathfrak{v}_\pi^\mu} + \left\langle \widehat{\mathcal{Q}}_i - \mathcal{Q}_i, \pi_i - \pi \right\rangle_{\mathfrak{v}_\pi^\mu}$$
$$= (1-\gamma)\left(\mathcal{V}_i(\mu) - \mathcal{V}_\pi(\mu)\right) + \left\langle \widehat{\mathcal{Q}}_i - \mathcal{Q}_i, \pi_i - \pi \right\rangle_{\mathfrak{v}_\pi^\mu}.$$

The term $\widehat{\mathcal{Q}}_i - \mathcal{Q}_i$ is exactly what we will control with the TD analysis, and thus the mirror descent approach has neatly decoupled concerns into an actor term, and a critic term.

In order to apply the mirror descent framework, we need to choose a *mirror mapping*. Rather than using $\phi$, for technical reasons we bake the measure $\mathfrak{v}_\pi^\mu$ into the mirror mapping and corresponding dual objects (cf. Appendix B and the proof of Lemma 2.1). This may seem strange, but it does not change the induced policy (it scales the dual object *for each state* by a constant), and thus is a degree of freedom, and allows us to state guarantees for *all* possible starting distributions for free.

Our full mirror descent setup is detailed in Appendix B, but culminates in the following guarantee.

**Lemma 2.1.** *Consider step size $\theta > 0$, any reference policy $\pi$, any starting measure $\mu$, and two treatments of the error $\widehat{\mathcal{Q}}_i - \mathcal{Q}_i$.*

1. *(**Simplified bound.**) Define $C_i := \sup_{s,a} |\widehat{\mathcal{Q}}_i(s,a)|$ for all $i < t$. Then*

$$K_{\mathfrak{v}_\pi^\mu}(\pi, \pi_t) + \theta(1-\gamma)\sum_{i<t}\left(\mathcal{V}_\pi(\mu) - \mathcal{V}_i(\mu)\right) \le K_{\mathfrak{v}_\pi^\mu}(\pi, \pi_0) + \theta^2 \sum_{i<t} C_i^2$$
$$+ \theta \sum_{i<t}\left\langle \widehat{\mathcal{Q}}_i - \mathcal{Q}_i, \pi_i - \pi \right\rangle_{\mathfrak{v}_\pi^\mu}.$$

2. *(**Refined bound.**) Define $\hat{\epsilon}_i := \sup_{s,a} |\widehat{\mathcal{Q}}_i(s,a) - \mathcal{Q}_i(s,a)|$. Then*

$$K_{\mathfrak{v}_\pi^\mu}(\pi, \pi_t) + \theta(1-\gamma)\sum_{i<t}\left(\mathcal{V}_\pi(\mu) - \mathcal{V}_i(\mu)\right) \le K_{\mathfrak{v}_\pi^\mu}(\pi, \pi_0) + \frac{\theta}{1-\gamma}$$
$$+ \theta \sum_{i<t}\left(\frac{2\gamma\hat{\epsilon}_i}{1-\gamma} + \hat{\epsilon}_i + \hat{\epsilon}_{i+1}\right),$$

*and additionally $\mathcal{V}_i$ and $\widehat{\mathcal{Q}}_i$ are approximately monotone: for any state $s$ and action $a$,*

$$\mathcal{V}_{i+1}(s) \ge \mathcal{V}_i(s) - \frac{2\hat{\epsilon}_i}{1-\gamma} \qquad \text{and} \qquad \widehat{\mathcal{Q}}_{i+1}(s,a) \ge \widehat{\mathcal{Q}}_i(s,a) - \frac{2\gamma\hat{\epsilon}_i}{1-\gamma} - \hat{\epsilon}_i - \hat{\epsilon}_{i+1}.$$

**Remark 2.2** (Regarding the mirror descent setup, Lemma 2.1)**.**

1. **Two rates.** For the refined bound, it is most natural to set $\theta = 1$, which requires $\mathcal{O}(1/\epsilon)$ iterations to reach accuracy $\epsilon > 0$; by contrast, the simplified guarantee requires $\mathcal{O}(1/\epsilon^2)$ iterations for the same $\epsilon > 0$ with step size $\theta = 1/\sqrt{t}$. We use the simplified form to prove Theorem 1.4, since its TD error term is less stringent; indeed, the TD analysis we provide in Section 3 will not be able to give the uniform control needed for the refined bound. Still, we feel the refined bound is promising, and include it for sake of completeness, future work, and comparison to prior work.

2. **Comparison to standard rates.** Comparing the refined bound (with all $\hat{\epsilon}_i$ terms set to zero) to the standard NPG rate in the literature (Agarwal et al., 2021b), the rate is exactly recovered; as such, this mirror descent setup at the very least has not paid a price in rates.

3. **Implicit regularization term.** A conspicuous difference between these bounds and both the standard NPG bounds (Agarwal et al., 2021b, Theorem 5.3), but also many mirror descent treatments, is the term $K_{\mathfrak{v}_{\bar{\pi}}^{\mu}}(\pi, \pi_t)$; one could argue that this term is nonnegative and moreover we care more about the value function, so why not drop it, as is usual? It is precisely this term that gives our implicit regularization effect: instead, we can drop *the value function term* and uniformly upper bound the right hand side to get $K_{\mathfrak{v}_{\bar{\pi}}^{\mu}}(\overline{\pi}, \pi_t) \leq \ln k + 1/(1-\gamma)^2$, which is how we control the entropy of the policy path, and prove Theorem 1.4. $\diamondsuit$

## 3 SAMPLING TOOLS

Via Lemma 2.1 above, our mirror descent black box analysis gives us a KL bound and a value function bound: what remains, and is the job of this section, is to control the $Q$ function estimation error, namely terms of the form $\mathcal{Q}_i - \widehat{\mathcal{Q}}_i$.

Our analysis here has two parts. The first part, as follows immediately, is that any bounded KL ball in policy space has uniformly controlled mixing times; the second part, which comes shortly thereafter, is our TD guarantees.

**Lemma 3.1.** *Let policy $\tilde{\pi}$ be given, and suppose the induced transition kernel on states $P_{\tilde{\pi}}$ is irreducible and aperiodic (Levin et al., 2006, Section 1.3). Then $\tilde{\pi}$ has a stationary distribution $\mathfrak{s}_{\tilde{\pi}}$, and moreover for any $c > 0$ and any measure $\nu$ which is positive on all states and a corresponding set of policies*

$$\mathcal{P}_c := \left\{ \pi : K_\nu(\tilde{\pi}, \pi) \leq c \right\},$$

*there exist constants $C, m_1, m_2$ so that mixing is uniform over $\mathcal{P}_c$, meaning for any $t$, and any $\pi \in \mathcal{P}_c$ with induced transition probabilities $P_\pi$,*

$$\sup_s \|P_\pi^t(s, \cdot) - \mathfrak{s}_\pi\|_{\mathrm{TV}} \leq m_1 e^{-m_2 t},$$

*and for any state $s$ and any $\pi \in \mathcal{P}_c$, and any action $a$ with $\tilde{\pi}(s, a) > 0$,*

$$\frac{1}{C} \leq \frac{\tilde{\pi}(s,a)}{\pi(s,a)} \leq C \qquad and \qquad \frac{1}{C} \leq \frac{\mathfrak{s}_{\tilde{\pi}}(s)}{\mathfrak{s}_\pi(s)} \leq C.$$

**Remark 3.2** (Implicit vs explicit exploration). On the surface, Lemma 3.1 might seem quite nice. Worrying about it a little more, and especially after inspecting the proof, it is clear that the constants $C$, $m_1$, and $m_2$ can be quite bad. On the one hand, one may argue that this is inherent to implicit exploration, and something like $\epsilon$-greedy is preferable, as it arguably gives an explicit control on all these quantities.

Some aspects of this situation are unavoidable, however. Consider a *combination lock* MDP, where a precise, hard-to-find sequence of actions must be followed to arrive at some good reward. Suppose this sequence has length $n$ and we have a reference policy $\tilde{\pi}$ which takes each of these good actions with probability $1 - 1/n$, whereby the probability of the sequence is $(1 - 1/n)^n \approx 1/e$; a policy $\pi \in \mathcal{P}_c$ with $\pi(s,a)/\tilde{\pi}(s,a) \leq 1/2$ for all actions $a$ can drop the probability of this good sequence of actions all the way down to $1/2^n$! $\diamondsuit$

Next we present our TD analysis. As discussed in Section 1, by contrast with prior work, our TD method does not make use of any projections, and does not require eigenvalue assumptions. The following guarantee is specialized to Algorithm 1; it is a corollary of a more general TD guarantee, given in Appendix C, which is stated without reference to Algorithm 1, and can be applied in other settings.

**Lemma 3.3** (See also Lemma C.4). *Suppose the MDP and linear MDP assumptions (cf. Assumptions 1.1 and 1.3). Consider a policy $\pi_i$ in some iteration $i$ of Algorithm 1, and suppose there exist mixing constants $m \geq 1$ and $c > 0$ so that the induced transition kernel $P_{\pi_i}$ on $\mathcal{S}$ satisfies*

$$\sup_s \|P_{\pi_i}^t(s, \cdot) - \mathfrak{s}_{\pi_i}\|_{\mathrm{TV}} \leq me^{-ct}.$$

*Suppose the TD iterations $N$ and step size $\eta$ satisfy*

$$N \geq k, \qquad \eta \leq \frac{1}{400\sqrt{kN}}, \qquad where \; k = \left\lceil \frac{\ln N + \ln m}{c} \right\rceil.$$

*Then letting $\mathbb{E}_i$ denote expectation over the trajectory $(s_{i,j}, a_{i,j})_{j \leq N}$ and letting $\bar{U}_i$ denote the expected TD fixed point given in Lemma C.3 (which satisfies $\|\bar{U}_i\| \leq 2/(1-\gamma)$), the average TD iterate $\widehat{U}_i := \frac{1}{N} \sum_{j<N} U_{i,j}$ satisfies*

$$\mathbb{E}_i \left\| \widehat{U}_i - \bar{U}_i \right\|^2 + \eta N \mathbb{E}_i \mathbb{E}_{(s,a) \sim (\mathfrak{s}_\pi, \pi)} \left\langle sa^\intercal, \widehat{U}_i - \bar{U}_i \right\rangle^2 \leq \frac{54}{(1-\gamma)^2},$$

*where $\left\langle sa^\intercal, \widehat{U}_i - \bar{U}_i \right\rangle = s^\intercal \widehat{U}_i a - s^\intercal \bar{U}_i a = \widehat{\mathcal{Q}}_i(s,a) - \mathcal{Q}_i(s,a)$ for almost every $(s,a)$.*

The proof is intricate owing mainly to issues of statistical dependency. It is not merely an issue that the chain is not started from the stationary distribution; dropping the subscript $i$ for convenience and letting $x_{j+1}$ and $x_j$ denote the vectorized forms of $s_{j+1} a_{j+1}^\intercal$ and $s_j a_j^\intercal$, and similarly letting $u_j$ denote the vectorized form of $U_j$, notice that $x_{j+1}, x_j, u_j$ are all statistically dependent. Indeed, even if $x_j$ is sampled from the stationary distribution (which also means $x_{j+1}$ is distributed according to the stationary distribution as well), the *conditional* distribution of $x_{j+1}$ given $x_j$ is not the same as that of $x_j$! To deal with such issues, the proof chooses a very small step size which ensures the TD estimate evolves much more slowly than the mixing time of the chain. On a more technical level, whenever the proof encounters an inner product of the form $\langle x_j, u_j \rangle$, it introduces a gap and instead considers $\langle u_{j-k}, x_j \rangle$, where $u_{j-k} - u_j$ is small due to the small step size, and these two are nearly independent due to fast mixing and the corresponding choice of $k$.

A second component of the proof, which removes projection steps from prior work (Bhandari et al., 2018), is an *implicit bias of TD*, detailed as follows. Mirroring the MD statement in Lemma 2.1, the left hand side here has not only a $\widehat{\mathcal{Q}}_i - \mathcal{Q}_i$ term as promised, but also a norm control $\|\widehat{U}_i - \bar{U}_i\|^2$; in fact, this norm control holds for all intermediate TD iterations, and is used throughout the proof to control many error terms. Just like in the MD analysis, this term is an *implicit regularization*, and is how this work avoids the projection step needed in prior work (Bhandari et al., 2018).

All the pieces are now in place to sketch the proof of Theorem 1.4, which is presented in full in Appendix D. To start, instantiate Lemma 3.1 with KL divergence upper bound $\ln k + 1/(1-\gamma)^2$, which gives the various mixing constants used throughout the proof (which we need to instantiate now, before seeing the sequence of policies, to avoid any dependence). With that out of the way, consider some iteration $i$, and suppose that for all iterations $j < i$, we have a handle both on the TD error, and also a guarantee that we are in a small KL ball around $\overline{\pi}$ (specifically, of radius $\ln k + 1/(1-\gamma)^2$). The right hand side of the simplified mirror descent bound in Lemma 2.1 only needs a control on all previous TD errors, therefore it implies both a bound on $\sum_{j<i} \mathcal{V}_j(s)$ and on $K_{\mathfrak{v}_{\overline{\pi}}^s}(\overline{\pi}, \pi_i)$. But this KL control on $\pi_i$ means that the mixing and other constants we assumed at the start will hold for $\pi_i$, and thus we can invoke Lemma 3.3 to bound the error on $\widehat{\mathcal{Q}}_i - \mathcal{Q}_i$, which we will use in the next loop of the induction. In this way, the actor and critic analyses complement each other and work together in each step of the induction.

## 4 DISCUSSION AND OPEN PROBLEMS

This work, in contrast to prior work in natural actor-critic and natural policy gradient methods, dropped many assumptions from the analysis, and many components from the algorithms. The analysis was meant to be fairly general purpose and unoptimized. As such, there are many open problems.

**Implicit vs explicit regularization/exploration.** What are some situations where one is better than the other, and vice versa? The analysis here only says you can get away with doing everything implicitly, but not necessarily that this is the best option.

**More general settings.** The paper here is for linear MDPs, linear softmax policies, finite state and action spaces. How much does the implicit bias phenomenon (and this analysis) help in more general settings?

ACKNOWLEDGMENTS

MT thanks Alekh Agarwal, Nan Jiang, Haipeng Luo, Gergely Neu, and Tor Lattimore for valuable discussions, as well as the detailed comments from the ICLR 2022 reviewers. The authors are grateful to the NSF for support under grant IIS-1750051.

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

## A  Background proof: existence of $\overline{\pi}$

The only thing in this section is the expanded version of Lemma 1.2, namely giving the unique maximum entropy optimal policy, and some key properties.

**Lemma A.1.** *If $|\mathcal{S}| < \infty$, then there exists a unique maximum entropy optimal policy $\overline{\pi}$ and corresponding $\overline{\mathcal{Q}}$ and $\overline{\mathcal{A}}$ which satisfy the following properties.*

1. *For any state $s$, let $\mathcal{A}_s$ denote the set of actions taken by optimal policies. Define $\overline{\pi}(s, \cdot) := \text{Uniform}(\mathcal{A}_s)$, which is unique; then $\overline{\pi}$ is also an optimal policy, and let $\overline{\mathcal{A}}$ and $\overline{\mathcal{Q}}$ denote its advantage and Q functions.*

2. *For every state $s$ and every action $a$, then $\overline{\mathcal{Q}}(s, a) = \max_\pi \mathcal{Q}_\pi(s, a)$, where the maximum is taken over all policies. Moreover, $\max_{a \in \mathcal{A}_s} \overline{\mathcal{Q}}(s, a) > \max_{a \notin \mathcal{A}_s} \overline{\mathcal{Q}}(s, a)$.*

3. *$\overline{\pi} = \lim_{r \to \infty} \phi(r\overline{\mathcal{A}})$.*

4. *If there exists an irreducible optimal policy, then $\overline{\pi}$ is irreducible as well, and moreover has a stationary distribution $\mathfrak{s}_{\overline{\pi}}$.*

*Proof of Lemmas 1.2 and A.1.*  1. Let $(s_1, \ldots, s_{|\mathcal{S}|})$ denote any enumeration of the state space $\mathcal{S}$. This proof will inductively construct a sequence of optimal policies $(\overline{\pi}_i)_{i=0}^{|\mathcal{S}|}$, where each $\overline{\pi}_i$ is optimal and satisfies $\overline{\pi}_i(s_j, \cdot) = \text{Uniform}(\mathcal{A}_{s_j})$ for $j \leq i$. For the base case, let $\overline{\pi}_0$ denote any optimal policy, which satisfies the desired conditions since the indexing on states starts from 1. For the inductive step, define $\overline{\pi}_{i+1}(s, \cdot) = \overline{\pi}_i(s, \cdot)$ for $s \neq s_{i+1}$ and $\overline{\pi}_{i+1}(s_{i+1}, \cdot) = \text{Uniform}(\mathcal{A}_{s_{i+1}})$. By the performance difference lemma, for any state $s$,

$$\mathcal{V}_{\overline{\pi}_i}(s) - \mathcal{V}_{\overline{\pi}_{i+1}}(s) = \frac{1}{1 - \gamma} \mathbb{E}_{s' \sim \mathfrak{v}_{\overline{\pi}_{i+1}}^s} \left\langle \mathcal{Q}_{\overline{\pi}_i}(s', \cdot), \overline{\pi}_i(s', \cdot) - \overline{\pi}_{i+1}(s', \cdot) \right\rangle.$$

By construction, the inner product is 0 for any $s' \neq s_{i+1}$. When $s' = s_{i+1}$, since $\overline{\pi}_i$ is optimal, then $\mathcal{V}_{\overline{\pi}_i}(s'')$ must be optimal for every state $s''$, which in turn means $\overline{\mathcal{Q}}_{\overline{\pi}_i}(s', \cdot)$ must be maximized at each $a \in \mathcal{A}_s$, and therefore the inner product is 0 in this case as well. This completes the induction, and the desired claim follows by noting $\overline{\pi} = \overline{\pi}_{|\mathcal{S}|}$.

2. For any $\pi$ with corresponding Q function $\mathcal{Q}_\pi$ and value function $\mathcal{V}_\pi$, and any $(s, a)$, then

$$\overline{\mathcal{Q}}(s, a) - \mathcal{Q}_\pi(s, a) = \mathbb{E}_{r, s' \sim (s, a)} \left[ r(s, a) + \gamma \overline{\mathcal{V}}(s') - r(s, a) - \gamma \mathcal{V}_\pi(s') \right]$$
$$= \gamma \mathbb{E}_{s' \sim (s, a)} \left[ \overline{\mathcal{V}}(s') - \mathcal{V}_\pi(s') \right]$$
$$\geq 0.$$

It follows that $\overline{\mathcal{Q}}(s, a) \geq \sup_\pi \mathcal{Q}_\pi(s, a)$, and since $\overline{\mathcal{Q}} = \mathcal{Q}_{\overline{\pi}}$, then in fact $\overline{\mathcal{Q}}(s, a) = \max_\pi \mathcal{Q}_\pi(s, a)$.

3. By the previous point, for any state $s$ and any $a \in \mathcal{A}_s$, then $\overline{\mathcal{A}}(s, a) = \overline{\mathcal{Q}}(s, a) - \overline{\mathcal{V}}(s) = 0$ whereas for any $b \notin \mathcal{A}_s$, then $\overline{\mathcal{A}}(s, b) = \overline{\mathcal{Q}}(s, b) - \overline{\mathcal{V}}(s) \leq \max_{b \notin \mathcal{A}_s} \overline{\mathcal{Q}}(s, b) - \min_{a \in \mathcal{A}_s} \overline{\mathcal{Q}}(s, a) < 0$. It follows that

$$\lim_{r \to \infty} \phi(r\mathcal{A}(s, \cdot)) = \text{Uniform}(\mathcal{A}_s) = \overline{\pi}(s, \cdot).$$

4. Let $\pi$ denote an arbitrary irreducible optimal policy. Since $\overline{\pi}$ is uniform on the set of optimal actions in any state, then for any pair $(s, s')$ and time $t > 0$ with $P_\pi^t(s, s') > 0$, then $P_{\overline{\pi}}^t(s, s') > 0$ as well. Since $\pi$ is irreducible, this holds for all pairs $(s, s')$, which means $\overline{\pi}$ is irreducible as well (Levin et al., 2006, Proposition 1.14), and has a stationary distribution $\mathfrak{s}_{\overline{\pi}}$.

$\square$

## B  Full mirror descent setup and proofs

This section first gives a basic mirror descent setup. This characterization is somewhat standard (Bubeck, 2015), though written with extra flexibility and with equalities to preserve implicit biases terms, which are dropped in most treatments.

First, here is the basic notation (which, unlike the paper body, will allow a subscripted step size $\theta_i$ which can differ between iterations):

$$
\begin{aligned}
p_{i+1} &:= p_i - \theta_i g_i, & \theta_i > 0, \\
q_i &:= \nabla \psi(p_i), & \text{closed proper convex } \psi, \\
\langle\!\langle \cdot, \cdot \rangle\!\rangle, & & \text{bilinear pairing}, \\
D(p, p_i) &:= \psi(p) - \left[ \psi(p_i) + \langle\!\langle q_i, p - p_i \rangle\!\rangle \right], & \text{primal Bregman divergence}, \\
D_*(q, q_i) &:= \psi^*(q) - \left[ \psi^*(q_i) + \langle\!\langle p_i, q - q_i \rangle\!\rangle \right], & \text{dual Bregman divergence}.
\end{aligned}
$$

One nonstandard choice here is that the Bregman divergence bakes in a conjugate element, rather than using $\nabla \psi$ and $\nabla \psi^*$; this gives an easy way to handle certain settings (like the boundary of the simplex) which run into non-uniqueness issues. Secondly, $\langle\!\langle \cdot, \cdot \rangle\!\rangle$ is just a bilinear form, and does not need to be interpreted as a standard inner product.

The standard Bregman identities used in mirror descent proofs are as follows:

$$
D_*(q, q_i) - D_*(q, q_{i+1}) - D_*(q_{i+1}, q_i) = \langle\!\langle p_i - p_{i+1}, q_{i+1} - q \rangle\!\rangle, \tag{B.1}
$$
$$
D_*(q_{i+1}, q_i) = D(p_i, p_{i+1}), \tag{B.2}
$$
$$
D_*(q_{i+1}, q_i) + D_*(q_i, q_{i+1}) = \langle\!\langle p_i - p_{i+1}, q_i - q_{i+1} \rangle\!\rangle, \tag{B.3}
$$
$$
D_*(q, q_i) = \psi^*(q) + \psi(p_i) - \langle\!\langle p_i, q \rangle\!\rangle \geq 0. \tag{B.4}
$$

With these in hand, the core mirror descent guarantee is as follows. The bound is written with equalities to allow for careful handling of error terms. Note that this version of mirror descent does not interpret the "gradient" $g_i$ in any way, and treats it as a vector and no more.

**Lemma B.1.** *Suppose $\theta_i > 0$. For any $t$ and $q$ where $q \in \mathrm{dom}(\psi^*)$,*

$$
\begin{aligned}
\sum_{i<t} \theta_i \langle\!\langle g_i, q_i - q \rangle\!\rangle &= D_*(q, q_0) - D_*(q, q_t) + \sum_{i<t} D(p_{i+1}, p_i) \\
&= D_*(q, q_0) - D_*(q, q_t) + \sum_{i<t} \left[ \langle\!\langle p_i - p_{i+1}, q_i - q_{i+1} \rangle\!\rangle - D_*(q_{i+1}, q_i) \right] \\
&= D_*(q, q_0) + \theta_0 \langle\!\langle g_0, q_0 \rangle\!\rangle - D_*(q, q_t) - \theta_{t-1} \langle\!\langle g_t, q_t \rangle\!\rangle - \sum_{i<t} D_*(q_{i+1}, q_i) \\
&\quad + \sum_{i=1}^{t-1} \langle\!\langle g_i, q_i \rangle\!\rangle (\theta_i - \theta_{i-1}) + \sum_{i<t} \theta_i \langle\!\langle g_{i+1} - g_i, q_{i+1} \rangle\!\rangle.
\end{aligned}
$$

*Moreover, for any $i$, $\langle\!\langle g_i, q_{i+1} \rangle\!\rangle \leq \langle\!\langle g_i, q_i \rangle\!\rangle$.*

*Proof.* For any fixed iterate $i < t$, by eqs. (B.1) to (B.3),

$$
\begin{aligned}
\theta_i \langle\!\langle g_i, q_i - q \rangle\!\rangle &= \langle\!\langle p_i - p_{i+1}, q_i - q \rangle\!\rangle \\
&= \langle\!\langle p_i - p_{i+1}, q_i - q_{i+1} \rangle\!\rangle + \langle\!\langle p_i - p_{i+1}, q_{i+1} - q \rangle\!\rangle \\
&= \langle\!\langle p_i - p_{i+1}, q_i - q_{i+1} \rangle\!\rangle + D_*(q, q_i) - D_*(q, q_{i+1}) - D_*(q_{i+1}, q_i) & \because \text{eq. (B.1)} \\
&= D_*(q, q_i) - D_*(q, q_{i+1}) + D_*(q_i, q_{i+1}) & \because \text{eq. (B.3)} \\
&= D_*(q, q_i) - D_*(q, q_{i+1}) + D(p_{i+1}, p_i), & \because \text{eq. (B.2)}
\end{aligned}
$$

and additionally note

$$
\frac{1}{\theta_i} \langle\!\langle p_i - p_{i+1}, q_i - q_{i+1} \rangle\!\rangle = \langle\!\langle g_i, q_i - q_{i+1} \rangle\!\rangle = \langle\!\langle g_{i+1} - g_i, q_{i+1} \rangle\!\rangle - \langle\!\langle g_{i+1}, q_{i+1} \rangle\!\rangle + \langle\!\langle g_i, q_i \rangle\!\rangle.
$$

The first equalities now follow by applying $\sum_{i<t}$ to both sides, telescoping, and using the various earlier Bregman identities (cf. eqs. (B.1) to (B.4)).

For the second part, for any $i$, by convexity of $\psi$,

$$0 \leq \langle\!\langle p_i - p_{i+1}, \nabla\psi(p_i) - \nabla\psi(p_{i+1}) \rangle\!\rangle = \theta_i \langle\!\langle g_i, q_i - q_{i+1} \rangle\!\rangle ,$$

which rearranges to give $\langle\!\langle g_i, q_{i+1} \rangle\!\rangle \leq \langle\!\langle g_i, q_i \rangle\!\rangle$ since $\theta_i > 0$. $\qquad\square$

All that remains is to instantiate the various mirror descent objects to match Algorithm 1, and control the resulting terms. This culminates in Lemma 2.1; its proof is as follows.

*Proof of Lemma 2.1.* The core of both parts of the proof is to apply the mirror descent guarantees from Lemma B.1, using the following choices. To start, the primal update is given by

$$g_i := -\widehat{\mathcal{Q}}_i,$$
$$p_{i+1} := p_i - \theta g_i = p_i + \theta \widehat{\mathcal{Q}}_i.$$

The mirror mapping and corresponding dual variables (with $\mathfrak{v}_\pi^\mu$ baked in) are

$$\psi(p) := \mathbb{E}_{s \sim \mathfrak{v}_\pi^\mu} \ln \sum_{a \in \mathcal{A}} \exp(p(s,a)) = \sum_{s \in \mathcal{S}} \mathfrak{v}_\pi^\mu(s) \ln \sum_{a \in \mathcal{A}} \exp(p(s,a)),$$

$$[\nabla\psi(p)](s,a) = \frac{\mathfrak{v}_\pi^\mu(s) \exp(p(s,a))}{\sum_{b \in \mathcal{A}} \exp(p(s,b))},$$

$$q_i := \nabla\psi(p_i),$$
$$q_i(s,a) := \mathfrak{v}_\pi^\mu(s)\pi_i(s,a).$$

The primal Bregman divergence, which uses a dual iterate rather than the mirror map, is

$$D(p, p_i) := \psi(p) - \left[ \psi(p_i) - \langle\!\langle q_i, p - p_i \rangle\!\rangle \right].$$

The inner product is the standard one, meaning

$$\langle\!\langle p, q \rangle\!\rangle := \langle p, q \rangle = \sum_{s \in \mathcal{S}} \sum_{a \in \mathcal{A}} p(s,a)q(s,a).$$

Lastly, the dual Bregman divergence is given by

$$\psi^*(q) = \begin{cases} \langle\!\langle q, \ln q \rangle\!\rangle - \sum_s \mathfrak{v}_\pi^\mu(s) \ln \mathfrak{v}_\pi^\mu(s), & q \in \Delta_{\mathcal{S} \times \mathcal{A}}, \\ \infty & \text{o.w.,} \end{cases}$$

$$D_*(q, q_i) := \psi^*(q) - \left[ \psi^*(q_i) - \langle\!\langle p_i, q - q_i \rangle\!\rangle \right]$$
$$= \langle\!\langle q, \ln q \rangle\!\rangle - \langle\!\langle q_i, \ln q_i \rangle\!\rangle$$
$$\quad - \sum_{s,a} \left( \ln(q_i(s,a)) + \ln \mathfrak{v}_\pi^\mu(s) + \ln \sum_b \exp(p_i(s,b)) \right) (q(s,a) - q_i(s,a))$$
$$= \left\langle\!\!\left\langle q, \ln \frac{q}{q_i} \right\rangle\!\!\right\rangle = K_{\mathfrak{v}_\pi^\mu}(\pi, \pi_i).$$

A key consequence of these constructions is that $q_i$, treated for any fixed $s$ as an unnormalized policy, agrees with $\pi_i := \phi(p_i)$ after normalization; that is to say, it gives the same policy, and the choice of $\mathfrak{v}_\pi^\mu$ baked into the definition is not needed by the algorithm, is only used in the analysis; the "gradient" $g_i = -\widehat{\mathcal{Q}}_i$ makes no use of it.

Plugging this notation in to Lemma B.1 but making use of two of its equalities, and the performance difference lemma (cf. eq. (1.2)), then for any $\mu$,

$$\theta(1-\gamma) \sum_{i<t} \left( \mathcal{V}_i(\mu) - \mathcal{V}_\pi(\mu) \right) = \sum_{i<t} \theta \langle \mathcal{Q}_i, \pi_i - \pi \rangle_{\mathfrak{v}_\pi^\mu}$$

$$= \sum_{i<t} \theta \left\langle \mathcal{Q}_i - \widehat{\mathcal{Q}}_i, \pi_i - \pi \right\rangle_{\mathfrak{v}_\pi^\mu} + \sum_{i<t} \left\langle \theta\widehat{\mathcal{Q}}_i, \pi_i - \pi \right\rangle_{\mathfrak{v}_\pi^\mu}$$

$$= \sum_{i<t} \theta \left\langle \mathcal{Q}_i - \widehat{\mathcal{Q}}_i, \pi_i - \pi \right\rangle_{\mathfrak{v}_\pi^\mu} - \sum_{i<t} \langle\!\langle \theta_i g_i, q_i - q_\pi \rangle\!\rangle . \qquad \text{(B.5)}$$

The proof now splits into the two different settings.

1. **(Simplified bound.)** By the above definitions and the first equality in Lemma B.1,
$$-\sum_{i<t} \langle\!\langle \theta_i g_i, q_i - q_\pi \rangle\!\rangle = D_*(q, q_t) - D_*(q, q_0) - \sum_{i<t} D(p_{i+1}, p_i),$$
where the last term may be bounded in a way common in the online learning literature (Shalev-Shwartz, 2011): since $e^z \leq 1 + z + z^2$ when $z \leq 1$, setting $Z(s, a) := \widehat{\mathcal{Q}}_i(s, a) - C_i$ for convenience (whereby $Z(s, a) \leq 0 \leq 1$ as needed by the preceding inequality),

$D(p_{i+1}, p_i)$

$$= \mathbb{E}_{s\sim\mathfrak{v}_\pi^\mu} \left( \ln \sum_a \exp(p_{i+1}(s, a)) - \ln \sum_a \exp(p_i(s, a)) - \sum_a \pi_i(s, a)(p_{i+1}(s, a) - p_i(s, a)) \right)$$

$$= \mathbb{E}_{s\sim\mathfrak{v}_\pi^\mu} \left( \ln \left( \sum_a \pi_i(s, a) \exp\left( \theta \widehat{\mathcal{Q}}_i(s, a) \right) \right) - \theta \sum_a \pi_i(s, a) \widehat{\mathcal{Q}}_i(s, a) \right)$$

$$= \mathbb{E}_{s\sim\mathfrak{v}_\pi^\mu} \left( \ln \left( \sum_a \pi_i(s, a) \exp\left( \theta Z(s, a) \right) \right) - \theta \sum_a \pi_i(s, a) Z(s, a) \right)$$

$$\leq \mathbb{E}_{s\sim\mathfrak{v}_\pi^\mu} \left( \ln \left( \sum_a \pi_i(s, a)(1 + \theta Z(s, a) + \theta^2 Z(s, a)^2) \right) - \theta \sum_a \pi_i(s, a) Z(s, a) \right)$$

$$\leq \mathbb{E}_{s\sim\mathfrak{v}_\pi^\mu} \left( \sum_a \pi_i(s, a)(1 + \theta Z(s, a) + \theta^2 Z(s, a)^2) - 1 - \theta \sum_a \pi_i(s, a) Z(s, a) \right)$$

$$= \mathbb{E}_{s\sim\mathfrak{v}_\pi^\mu} \sum_a \pi_i(s, a) \theta^2 Z(s, a)^2 \leq \theta^2 C_i^2,$$

which together with the preceding as well as eq. (B.5) gives

$$\theta(1 - \gamma) \sum_{i<t} \left( \mathcal{V}_i(\mu) - \mathcal{V}_\pi(\mu) \right)$$

$$= \sum_{i<t} \theta \left\langle \mathcal{Q}_i - \widehat{\mathcal{Q}}_i, \pi_i - \pi \right\rangle_{\mathfrak{v}_\pi^\mu} - \sum_{i<t} \langle\!\langle \theta_i g_i, q_i - q_\pi \rangle\!\rangle .$$

$$\geq \sum_{i<t} \theta \left\langle \mathcal{Q}_i - \widehat{\mathcal{Q}}_i, \pi_i - \pi \right\rangle_{\mathfrak{v}_\pi^\mu} + K_{\mathfrak{v}_\pi^\mu}(\pi, \pi_t) - K_{\mathfrak{v}_\pi^\mu}(\pi, \pi_0) - \theta^2 \sum_{i<t} C_i^2,$$

which gives the desired bound after rearranging.

2. **(Refined bound.)** By Lemma B.1 with the above choices and any measure $\mathfrak{v}_\pi^\mu$, then
$$\left\langle \widehat{\mathcal{Q}}_i, \pi_{i+1} \right\rangle_{\mathfrak{v}_\pi^\mu} = - \langle\!\langle g_i, q_{i+1} \rangle\!\rangle \geq - \langle\!\langle g_i, q_i \rangle\!\rangle = \left\langle \widehat{\mathcal{Q}}_i, \pi_i \right\rangle_{\mathfrak{v}_\pi^\mu}, \tag{B.6}$$
and also
$$-\sum_{i<t} \langle\!\langle \theta_i g_i, q_i - q_\pi \rangle\!\rangle = -D_*(q, q_0) - \theta \langle\!\langle g_0, q_0 \rangle\!\rangle + D_*(q, q_t) + \theta \langle\!\langle g_t, q_t \rangle\!\rangle + \sum_{i<t} D_*(q_{i+1}, q_i)$$

$$- \sum_{i<t} \theta \langle\!\langle g_{i+1} - g_i, q_{i+1} \rangle\!\rangle$$

$$\geq K_{\mathfrak{v}_\pi^\mu}(\pi, \pi_t) - K_{\mathfrak{v}_\pi^\mu}(\pi, \pi_0) - \frac{\theta}{1 - \gamma} + \sum_{i<t} \theta \left\langle \widehat{\mathcal{Q}}_{i+1} - \widehat{\mathcal{Q}}_i, \pi_{i+1} \right\rangle_{\mathfrak{v}_\pi^\mu}.$$

To simplify these further, first note by eq. (B.6) with measure $\mathfrak{v}_{\pi_{i+1}}^{\delta_s}$ for any state $s$ and the performance difference lemma that

$$0 \leq \left\langle \widehat{\mathcal{Q}}_i, \pi_{i+1} - \pi_i \right\rangle_{\mathfrak{v}_{\pi_{i+1}}^{\delta_s}}$$

$$= \left\langle \widehat{\mathcal{Q}}_i - \mathcal{Q}_i, \pi_{i+1} - \pi_i \right\rangle_{\mathfrak{v}_{\pi_{i+1}}^{\delta_s}} + \left\langle \mathcal{Q}_i, \pi_{i+1} - \pi_i \right\rangle_{\mathfrak{v}_{\pi_{i+1}}^{\delta_s}}$$

$$\leq 2\hat{\epsilon}_i + (1 - \gamma) \left( \mathcal{V}_{i+1}(\delta_s) - \mathcal{V}_i(\delta_s) \right),$$

which rearranges to give

$$\mathcal{V}_{i+1}(\delta_s) \geq \mathcal{V}_i(\delta_s) - \frac{2\hat{\epsilon}_i}{1-\gamma} \qquad \forall s,$$

which itself in turn implies

$$\left\langle \widehat{\mathcal{Q}}_{i+1} - \widehat{\mathcal{Q}}_i, \pi_{i+1} \right\rangle_{\mathfrak{v}_\pi^\mu} \geq \left\langle \mathcal{Q}_{i+1} - \mathcal{Q}_i, \pi_{i+1} \right\rangle_{\mathfrak{v}_\pi^\mu} - \hat{\epsilon}_i - \hat{\epsilon}_{i+1}$$

$$= \gamma \mathbb{E}_{\substack{s \sim \mathfrak{v}_\pi^\mu \\ a \sim \pi_{i+1}(s,\cdot) \\ s' \sim (s,a)}} \left( \mathcal{V}_{i+1}(\delta_{s'}) - \mathcal{V}_i(\delta_{s'}) \right) - \hat{\epsilon}_i - \hat{\epsilon}_{i+1}$$

$$\geq -\frac{2\gamma\hat{\epsilon}_i}{1-\gamma} - \hat{\epsilon}_i - \hat{\epsilon}_{i+1}.$$

(The preceding derivation works if $\mathfrak{v}_\pi^\mu$ is replaced with any measure on states.) Plugging this all back in to eq. (B.5) gives

$$\mathcal{V}_{t-1}(\mu) - \mathcal{V}_\pi(\mu) \geq \frac{1}{t} \sum_{i<t} \left( \mathcal{V}_i(\mu) - \mathcal{V}_\pi(\mu) \right) - \sum_{i<t} \frac{2(1+i)\epsilon_i}{t(1-\gamma)}$$

$$\geq \frac{K_{\mathfrak{v}_\pi^\mu}(\pi, \pi_t) - K_{\mathfrak{v}_\pi^\mu}(\pi, \pi_0)}{t\theta(1-\gamma)} - \frac{1}{t(1-\gamma)^2} - \frac{1}{t(1-\gamma)} \sum_{i<t} \left( \frac{2\gamma\epsilon_i}{1-\gamma} + \epsilon_i + \epsilon_{i+1} \right)$$

$$- \sum_{i<t} \frac{2(1+i)\epsilon_i}{t(1-\gamma)},$$

where the last term may be omitted for the summation version, and these expressions rearrange to give the final bounds.

$\square$

## C  SAMPLING PROOFS

As in the body, this section both provides tools to control mixing times, and also a generalized TD analysis.

### C.1  MIXING TIME CONTROLS WITHIN KL BALLS

To control mixing times, these lemmas will use the notion of *conductance*:

$$\Phi_\pi^* := \min_{S \subseteq |\mathcal{S}| : \mathfrak{s}_\pi(S) \leq 1/2} \Phi_\pi(S), \qquad \text{where } \Phi_\pi(S) := \frac{\sum_{s \in S} \sum_{s' \notin S} \mathfrak{s}_\pi(s) P_\pi(s, s')}{\mathfrak{s}_\pi(S)}.$$

This quantity was shown to control mixing times of reversible chains by Jerrum & Sinclair (1988) (for more discussion, see Levin et al. (2006, eq. (7.7), Theorem 12.4, Theorem 13.10)), but a later proof due to Lovasz & Simonovits (1990) requires chains to only be lazy, and does not need reversibility. Our chains can be made lazy by flipping a coin before each step, but in our setting we can avoid this and directly use the mixing time of the lazy chains to control the mixing time of the original chains.

As a first tool, note that if policies are similar, then their stationary distributions and conductances are also similar.

**Lemma C.1.** *Let constant $c > 0$ and an aperiodic irreducible policy $\tilde{\pi}$ be given, and define a set of policies whose action probabilities are similar:*

$$\mathcal{P} := \left\{ \pi \ : \ \forall s, a, \tilde{\pi}(s,a) > 0 \centerdot \frac{1}{c} \leq \frac{\pi(s,a)}{\tilde{\pi}(s,a)} \leq c \right\}.$$

*Then there exists a constant $C > 0$ so that the stationary distributions and conductances are also similar: for any $\pi \in \mathcal{P}$ and any state $s$,*

$$\frac{1}{C} \leq \frac{\mathfrak{s}_\pi(s,a)}{\mathfrak{s}_{\tilde{\pi}}(s,a)} \leq C, \qquad \Phi_\pi^* \geq \frac{1}{C} \Phi_{\tilde{\pi}}^*.$$

*Proof.* First note that since $\tilde{\pi}$ is aperiodic and irreducible, it has a stationary distribution $\mathfrak{s}_{\tilde{\pi}}$ where necessarily $\mathfrak{s}_{\tilde{\pi}}(s) > 0$ for all states $s$. Next, recall that the stationary distribution can be characterized in terms of hitting times (Levin et al., 2006, Proposition 1.19), meaning

$$\mathfrak{s}_{\tilde{\pi}}(s) = \frac{1}{\mathbb{E}[\min\{t > 0 : s_t = s\}|s_0 = s]}.$$

As discussed in proofs that the denominator is finite (Levin et al., 2006, proof of Lemma 1.13), letting $P_\pi$ and $P_{\tilde{\pi}}$ corresponding to the state transitions induced by taking a step by $\pi$ and $\tilde{\pi}$ respectively and then using the MDP dynamics to get to a state, there exist $r > 0$ and $\epsilon > 0$ so that $P_{\tilde{\pi}}^j(s, s') > \epsilon$ for any states $s, s'$ and any $j \geq r$, therefore $P_{\tilde{\pi}}^j(s, s') \geq \epsilon c^{-j}$. In fact, for a fixed $s$, letting $j_s \geq 1$ denote the smallest exponent so that $P_{\tilde{\pi}}^{j_s}(s, s) > 0$, then $\mathbb{E}_{\tilde{\pi}}[\min\{t > 0 : s_t = s\}|s_0 = s] \geq j_s$, meanwhile, as in Levin et al. (2006, proof of Lemma 1.13), defining $\tau_\pi^+ := \min\{t > 0 : s_t = s\}$,

$$
\begin{aligned}
\mathbb{E}[\tau_\pi^+ | s_0 = s] &= \sum_{t \geq 0} \Pr[\tau_\pi^+ > t | s_0 = s] \\
&\leq \sum_{k \geq 0} j_s \Pr[\tau_\pi^+ > k j_s | s_0 = s] \\
&\leq j_s \sum_{k \geq 0} (1 - c^{-j_s} P_{\tilde{\pi}}^{j_s}(s, s))^k \\
&\leq \frac{j_s}{c^{-j_s} P_{\tilde{\pi}}^{j_s}(s, s)} \\
&\leq \frac{\mathbb{E}[\tau_{\tilde{\pi}}^+ | s_0 = s]}{c^{-j_s} P_{\tilde{\pi}}^{j_s}(s, s)},
\end{aligned}
$$

where the denominator does not depend on $\pi$ and thus the ratio is uniformly bounded over $\mathcal{P}$. (We can produce a bound in the reverse direction trivially, by using $\mathfrak{s}_\pi(s)/\mathfrak{s}_{\tilde{\pi}}(s) \leq 1/\mathfrak{s}_{\tilde{\pi}}(s)$.) Let $C_0$ denote the maximum of this ratio and $c$.

Bounding the conductance is an easy consequence of the definition of $C_0$: using $C_0$ to swap various terms depending on $\pi$ with terms depending on $\tilde{\pi}$, it follows that

$$\Phi_\pi^* \geq \frac{1}{C_0^4} \Phi_{\tilde{\pi}}^*.$$

The proof is now complete by taking $C := C_0^4$ as the chosen constant. $\qquad\square$

Next, we use the preceding fact to obtain mixing times; the proof will need to convert to lazy chains to invoke the mixing time bound due to Lovasz & Simonovits (1990), but then will use a characterization of mixing times via coupling times to reason about the original chain (Levin et al., 2006, Theorem 5.4).

**Lemma C.2.** *Let constant $c > 0$ and an aperiodic irreducible policy $\tilde{\pi}$ be given, and define a set of policies*

$$\mathcal{P} := \left\{ \pi \; : \; \forall s, a, \tilde{\pi}(s, a) > 0 \bullet \frac{1}{c} \leq \frac{\pi(s, a)}{\tilde{\pi}(s, a)} \leq c \right\}.$$

*Then there exist constants $m_1, m_2 > 0$ so that for every $\pi \in \mathcal{P}$ and every $t$,*

$$\sup_s \|P_\pi^t(s, \cdot) - \mathfrak{s}_\pi\|_{\mathrm{TV}} \leq m_1 e^{-m_2 t}.$$

*Proof.* For any policy $\pi$, let $\nu_\pi$ denote the corresponding *lazy chain*: that is, in any state, $\nu_\pi$ does nothing with probability $1/2$ (it stays in the same state but does not interact with the MDP to receive any reward or transition), and otherwise with probability $1/2$ uses $\pi$ to interact with the MDP in its current state. Equivalently, if $s \neq s'$, then $P_{\nu_\pi}(s, s') = P_\pi(s, s')/2$, whereas $P_{\nu_\pi}(s, s) = (1 + P_\pi(s, s))/2$. Define $\tilde{\nu} := \nu_{\tilde{\pi}}$ for convenience, and a new set of nearby policies:

$$\mathcal{P}_\nu := \left\{ \pi \; : \; \forall s, s', P_{\tilde{\nu}}(s, s') > 0 \bullet \frac{1}{c'} \leq \frac{P_{\nu_\pi}(s, s')}{P_{\tilde{\nu}}(s, s')} \leq c' \right\},$$

where it will be shown that $c'$ can be chosen so that $\mathcal{P}_\nu \supseteq \mathcal{P}$. Indeed, by definition of $\mathcal{P}$, since $\pi/\tilde{\pi}$ is bounded, then so is $P_\pi/P_{\tilde{\pi}}$, and in particular pick any $c' > 0$ so that for any $\pi$ and any pair of states $(s, s')$ with $P_{\tilde{\pi}}(s, s') > 0$, it holds that

$$\frac{1}{c'} \leq \frac{P_\pi(s, s')}{P_{\tilde{\pi}}(s, s')} \leq c'.$$

To show that this choice of $c'$ suffices, first consider the case of a pair of different states states $s \neq s'$: then

$$\frac{1}{c'} \leq \frac{P_\pi(s, s')}{P_{\tilde{\pi}}(s, s')} = \frac{2P_{\nu_\pi}(s, s')}{2P_{\tilde{\nu}}(s, s')} = \frac{P_{\nu_\pi}(s, s')}{P_{\tilde{\nu}}(s, s')} = \frac{P_\pi(s, s')}{P_{\tilde{\pi}}(s, s')} \leq c';$$

whereas in the case $s = s'$, if $P_\pi(s, s) \geq P_{\tilde{\pi}}(s, s)$, then

$$c' \geq \frac{P_\pi(s, s)}{P_{\tilde{\pi}}(s, s)} \geq \frac{(1 + P_\pi(s, s))/2}{(1 + P_{\tilde{\pi}}(s, s))/2} = \frac{P_{\nu_\pi}(s, s)}{P_{\tilde{\nu}}(s, s)} \geq 1,$$

with an analogous relationship when $P_\pi(s, s) \leq P_{\tilde{\pi}}(s, s)$, which implies

$$c' \geq \frac{P_{\nu_\pi}(s, s)}{P_{\tilde{\nu}}(s, s)} \geq \frac{1}{c'}.$$

Consequently, it follows that $\mathcal{P}_\nu \supseteq \mathcal{P}$, and by Lemma C.1, that there exists a constant $C$ so that every $\pi \in \mathcal{P}$ satisfies

$$\frac{1}{C} \leq \frac{\mathfrak{s}_{\nu_\pi}(s, a)}{\mathfrak{s}_{\tilde{\nu}}(s, a)} \leq C, \qquad \Phi_{\nu_\pi}^* \geq \frac{1}{C} \Phi_{\tilde{\nu}}^*.$$

(One reason for the prevalence of lazy chains is that they always have unique stationary distributions (see, e.g., (Levin et al., 2006; Lovasz & Simonovits, 1990)), but as in the preceding, in our setting we always have stationary distributions automatically; thus we did not need to use laziness algorithmically, and can use it analytically.)

Since the conductance is uniformly bounded for every element of $\mathcal{P}_\nu$, there exist positive constants $m_3, m_4$ so that for every $\pi \in \mathcal{P}_\nu$, (Lovasz & Simonovits, 1990),

$$\sup_s \|P_{\nu_\pi}^t(s, \cdot) - \mathfrak{s}_{\nu_\pi}\|_{\mathrm{TV}} \leq m_3 \exp(-m_4 t).$$

Now define $p_{\min} := \min_s \mathfrak{s}_{\tilde{\nu}}(s)/C > 0$, whereby it holds by the above that $p_{\min} \leq \inf_{\pi \in \mathcal{P}_\nu} \min_s \mathfrak{s}_{\nu_\pi}(s)$. As such, by the preceding mixing bound, there exists a $t_0$ so that, simultaneously for every $\pi \in \mathcal{P}_\nu$, for all $t \geq t_0$, by the definition of total variation (instantiated on singletons), for every $s'$, $\sup_s P_{\nu_\pi}^t(s, s') \geq p_{\min}/2 > 0$. Now consider some fixed $\pi \in \mathcal{P}$, which also satisfies $\pi \in \mathcal{P}_\nu$ since $\mathcal{P}_\nu \supseteq \mathcal{P}$ as above, and consider the *coupling time* of two sequences $(x_0, x_1, \ldots)$ and $(y_0, y_1, \ldots)$ which start from some arbitrary pair of states $(x_0, y_0)$, and thereafter each steps according to $P_\pi$ (Levin et al., 2006, Section 5.2). Instead of running $P_\pi$ directly, simulate it as follows: use $P_{\nu_\pi}$, but discard from the sequence any steps which invoke the lazy option. Then, for $t \geq t_0$, the probability of both chains not choosing the lazy option and jumping to the same state is at least $p := \sum_s (p_{\min}/2)^2/4 > 0$. As such, for any $t \geq t_0$, the probability that the two chains did not land on the same state somewhere between $t_0$ and $t$ is at most

$$(1 - p)^{t - t_0} \leq (1 - p)^{-t_0} \exp(-pt).$$

But this is exactly an upper bound on the coupling time, meaning by (Levin et al., 2006, Theorem 5.4) that

$$\sup_{x_0, y_0} \|P_\pi^t(x_0, \cdot) - P_\pi^t(y_0, \cdot)\|_{\mathrm{TV}} \leq (1 - p)^{-t_0} \exp(-pt),$$

which in turn directly bounds the mixing time (Levin et al., 2006, Corollary 5.5), indeed with constants $m_1 := (1 - p)^{-t_0}$ and $m_2 = p$. Since these constants do not depend on the specific policy $\pi$ in any way, the mixing time has been uniformly controlled as desired. $\qquad \square$

With these tools in hand, we may now control mixing times uniformly over KL balls.

*Proof of Lemma 3.1.* By definition of $K_\nu$ and $\mathcal{P}_c$, for any $\pi \in \mathcal{P}_c$, letting $(s', a')$ denote any pair which maximizes $\tilde{\pi}(s, a)/\pi(s, a)$ (which implies $\tilde{\pi}(s', a') > 0$), and since $\sum_a q_a \ln q_a \geq -\ln k$ for any probability vector $q$ over actions,

$$
\begin{aligned}
c \geq K_\nu(\tilde{\pi}, \pi) &= \sum_{s \in \mathcal{S}} \nu(s) \sum_a \tilde{\pi}(s, a) \ln \frac{\tilde{\pi}(s, a)}{\pi(s, a)} \\
&\geq \nu(s') \tilde{\pi}(s', a') \ln \frac{\tilde{\pi}(s', a')}{\pi(s', a')} + \sum_{(s,a) \neq (s',a')} \nu(s) \tilde{\pi}(s, a) \ln \frac{\tilde{\pi}(s, a)}{\pi(s, a)} \\
&\geq \nu(s') \tilde{\pi}(s', a') \ln \frac{\tilde{\pi}(s', a')}{\pi(s', a')} + \nu(s') \sum_{a \neq a'} \tilde{\pi}(s', a) \ln \frac{\tilde{\pi}(s', a)}{\pi(s', a)} \\
&\geq \nu(s') \tilde{\pi}(s', a') \ln \frac{\tilde{\pi}(s', a')}{\pi(s', a')} - \nu(s') \ln k,
\end{aligned}
$$

then

$$
\begin{aligned}
\frac{\tilde{\pi}(s', a')}{\pi(s', a')} &\leq \exp \left( \frac{c}{\nu(s') \tilde{\pi}(s', a')} + \frac{\ln k}{\tilde{\pi}(s', a')} \right) \\
&\leq \max \left\{ \exp \left( \frac{c}{\tilde{\pi}(s'', a'') \min_s \nu(s)} + \frac{\ln k}{\tilde{\pi}(s'', a'')} \right) \ : \ \tilde{\pi}(s'', a'') > 0 \right\},
\end{aligned}
$$

where the final expression does not depend on $\pi$; it follows that $\tilde{\pi}(s, a)/\pi(s, a)$ is uniformly upper bounded over $\mathcal{P}_c$. On the other hand, $\tilde{\pi}(s, a)/\pi(s, a) \geq \tilde{\pi}(s, a)$, so the ratio uniformly bounded in both directions over $\mathcal{P}_c$, and let $C_0$ denote this ratio.

This in turn completes the proof: via Lemma C.1, the ratio of stationary distributions is also uniformly controlled, and via Lemma C.2, the mixing times are uniformly controlled. To obtain the final constants, it suffices to take the maximum of the relevant constants given by the preceding. $\square$

## C.2 TD GUARANTEES

The first step is to characterize the fixed points of the TD update, which in turn motivates the linear MDP assumption (cf. Assumption 1.3), and is fairly standard (Bhandari et al., 2018).

**Lemma C.3.** *Let any policy $\pi$ be given, and suppose $x \sim (\mathfrak{s}_\pi, \pi)$ is sampled from the stationary distribution (in vectorized state/action form), and $x'$ is a subsequent sample. Then, letting $(\mathbb{E} x x^\intercal)^+$ denote the pseudoinverse of $\mathbb{E} x x^\intercal$,*

$$
\bar{u} := \sum_{t \geq 0} \left( \gamma \left[ \mathbb{E} x x^\intercal \right]^+ \mathbb{E} x (x')^\intercal \right)^t \left[ \mathbb{E} x x^\intercal \right]^+ \mathbb{E} x r
$$

*is a fixed point of the expected TD update, meaning*

$$
\bar{u} = \mathbb{E} \left( \bar{u} - \eta x \left( \langle x - \gamma x', \bar{u} \rangle - r \right) \right).
$$

*Moreover, under the linear MDP assumption (cf. Assumption 1.3), then $x_{sa}^\intercal \bar{u} = \mathcal{Q}_\pi(s, a)$ for almost every $(s, a)$. Lastly, $\|\bar{u}\| \leq 2/(1 - \gamma)$.*

*Proof.* Let $\mathcal{X}$ denote the span of the support of $x$ according to its stationary distribution; since $\mathbb{E} x x^\intercal$ is symmetric and real, then $\mathbb{E} x x^\intercal (\mathbb{E} x x^\intercal)^+ = \Pi_\mathcal{X}$, where $\Pi_\mathcal{X}$ denotes orthogonal projection onto $\mathcal{X}$.

For the form of the fixed point, it suffices to show

$$
\mathbb{E} x \left( \langle x - \gamma x', \bar{u} \rangle - r \right) = 0.
$$

To this end, first note that

$$\left[\mathbb{E}xx^{\mathsf{T}}\right]\bar{u} = \left[\mathbb{E}xx^{\mathsf{T}}\right]\sum_{t\geq 0}\left(\gamma\left[\mathbb{E}xx^{\mathsf{T}}\right]^{+}\mathbb{E}x(x')^{\mathsf{T}}\right)^{t}\left[\mathbb{E}xx^{\mathsf{T}}\right]^{+}\mathbb{E}xr$$

$$= \left[\mathbb{E}xx^{\mathsf{T}}\right]\left[\mathbb{E}xx^{\mathsf{T}}\right]^{+}\mathbb{E}xr + \left[\mathbb{E}xx^{\mathsf{T}}\right]\sum_{t\geq 1}\left(\gamma\left[\mathbb{E}xx^{\mathsf{T}}\right]^{+}\mathbb{E}x(x')^{\mathsf{T}}\right)^{t}\left[\mathbb{E}xx^{\mathsf{T}}\right]^{+}\mathbb{E}xr$$

$$= \Pi_{\mathcal{X}}\mathbb{E}xr + \left[\mathbb{E}xx^{\mathsf{T}}\right]\gamma\left[\mathbb{E}xx^{\mathsf{T}}\right]^{+}\left[\mathbb{E}x(x')^{\mathsf{T}}\right]\sum_{t\geq 0}\left(\gamma\left[\mathbb{E}xx^{\mathsf{T}}\right]^{+}\mathbb{E}x(x')^{\mathsf{T}}\right)^{t}\left[\mathbb{E}xx^{\mathsf{T}}\right]^{+}\mathbb{E}xr$$

$$= \Pi_{\mathcal{X}}\mathbb{E}xr + \gamma\Pi_{\mathcal{X}}\left[\mathbb{E}x(x')^{\mathsf{T}}\right]\bar{u}.$$

$$= \mathbb{E}xr + \gamma\left[\mathbb{E}x(x')^{\mathsf{T}}\right]\bar{u}.$$

This completes the fixed point claim, since

$$\mathbb{E}x\left(\langle x - \gamma x, \bar{u}\rangle - r\right) = \left[\mathbb{E}xx^{\mathsf{T}}\right]\bar{u} - \gamma\left[\mathbb{E}x(x')^{\mathsf{T}}\right]\bar{u} - \mathbb{E}xr = 0.$$

For the claims under the linear MDP assumption, using the notation from Assumption 1.3, letting $X_{\pi}$ denote the transition mapping induced by $\pi$, note by the tower property that

$$\mathbb{E}\left(x(x')^{\mathsf{T}}\right) = \mathbb{E}\left(x\mathbb{E}\left[(x')^{\mathsf{T}}|x\right]\right) = \mathbb{E}\left(x\mathbb{E}\left[(X_{\pi}v')^{\mathsf{T}}|x\right]\right) = \mathbb{E}\left(x(X_{\pi}Mx)^{\mathsf{T}}\right) = \mathbb{E}xx^{\mathsf{T}}M^{\mathsf{T}}X_{\pi}^{\mathsf{T}},$$

and therefore, for any $x_{sa}$ in the support of $\pi$ (which implies $x_{sa}\in\mathcal{X}$), and since $x'\in\mathcal{X}$ as well,

$$x_{sa}^{\mathsf{T}}\bar{u} = x_{sa}^{\mathsf{T}}\sum_{t\geq 0}\left(\gamma\left[\mathbb{E}xx^{\mathsf{T}}\right]^{+}\mathbb{E}x(x')^{\mathsf{T}}\right)^{t}\left[\mathbb{E}xx^{\mathsf{T}}\right]^{+}\mathbb{E}xr$$

$$= x_{sa}^{\mathsf{T}}\sum_{t\geq 0}\left(\gamma\left[\mathbb{E}xx^{\mathsf{T}}\right]^{+}\mathbb{E}xx^{\mathsf{T}}M^{\mathsf{T}}X_{\pi}^{\mathsf{T}}\right)^{t}\left[\mathbb{E}xx^{\mathsf{T}}\right]^{+}\mathbb{E}xx^{\mathsf{T}}y$$

$$= x_{sa}^{\mathsf{T}}\sum_{t\geq 0}\left(\gamma\Pi_{\mathcal{X}}M^{\mathsf{T}}X_{\pi}^{\mathsf{T}}\right)^{t}\Pi_{\mathcal{X}}y$$

$$= x_{sa}^{\mathsf{T}}\sum_{t\geq 0}\left(\gamma M^{\mathsf{T}}X_{\pi}^{\mathsf{T}}\right)^{t}y$$

$$= \mathcal{Q}_{\pi}(s, a).$$

Lastly, since $\bar{u}\in\mathcal{X}$, and since $|\mathcal{Q}_{\pi}(s,a)|\leq 1/(1-\gamma)$ due to rewards lying within $[0,1]$, and since $1/2\leq\|s\|\leq 1$, then

$$\|\bar{u}\| = \sup_{x_{sa}\in\mathcal{X}}\frac{|\bar{u}^{\mathsf{T}}x_{sa}|}{\|s\|} = \sup_{x_{sa}\in\mathcal{X}}\frac{\mathcal{Q}_{\pi}(s,a)}{\|s\|} \leq \frac{2}{1-\gamma}.$$

$\square$

Now comes the core TD guarantee. In comparison with prior work (Bhandari et al., 2018), the need for projections and starting from the stationary distribution are both dropped.

**Lemma C.4.** *Let a policy $\pi$ be given which interacts with an MDP whose states $s\in\mathbb{R}^{d}$ satisfy $\|s\|\leq 1$, and whose action set is finite and represented as $\{e_{1},\ldots,e_{k}\}$; for convenience, let $x = \mathrm{vec}(se_{k}^{\mathsf{T}})$ denote a canonical vectorization of state/action pairs, as in the body of the paper. Let $P_{\pi}$ be the Markov chain on states induced by policy $\pi$, and assume it satisfies the following mixing time bound:*

$$\sup_{s}\|P_{\pi}^{t}(s,\cdot) - \mathfrak{s}_{\pi}\|_{\mathrm{TV}} \leq me^{-ct}.$$

*Let state/action/reward triples $(s_{j}, a_{j}, r_{j})_{j<N}$ be sampled via interaction of $\pi$ with the MDP, where the initial state $s_{0}$ is arbitrary, the random rewards satisfy $r_{j}\in[0,1]$ almost surely, and write $x_{j} = \mathrm{vec}(s_{j}a_{j}^{\mathsf{T}})$, and let $\mathbb{E}_{\vec{x},\vec{r}}$ denote the expectation over this trajectory.*

*Consider the stochastic TD updates defined recursively via $u_{0} = 0$ and thereafter*

$$u_{j+1} := u_{j} - \eta x_{j}\left(u^{\mathsf{T}}x_{j} - \gamma u^{\mathsf{T}}x_{j+1} - r_{j}\right),$$

*and let $\bar{u}$ be a fixed point of the corresponding expected TD updates with stationary samples, meaning*

$$\mathbb{E}_{\substack{x,r\sim(\mathfrak{s}_\pi,\pi)\\x'\sim x}} x\left(\left\langle x - \gamma x', \bar{u}\right\rangle - r\right) = 0,$$

*where $x$ is sampled from the stationary distribution and $x'$ is a subsequent sample, and assume $\|\bar{u}\| \leq 2/(1-\gamma)$.*

*If the TD parameters $N$ and $\eta$ are chosen according to*

$$N \geq k, \qquad \eta \leq \frac{1}{400\sqrt{kN}}, \qquad \text{where } k = \left\lceil \frac{\ln N + \ln m}{c} \right\rceil,$$

*then the average TD iterate $\hat{u} := \frac{1}{N}\sum_{j<N} u_j$ satisfies*

$$\mathbb{E}_{\vec{x},\vec{r}}\left(\|\hat{u} - \bar{u}\|^2 + \eta N \mathbb{E}_{x_{sa}\sim(\mathfrak{s}_\pi,\pi)}\left\langle x_{sa}, \hat{u} - \bar{u}\right\rangle^2\right) \leq \frac{54}{(1-\gamma)^2}.$$

*Proof.* The structure and primary concerns of the proof are as follows. The main issue is that $u_j, x_j, x_{j+1}$ are statistically dependent; in fact, even in the unlikely but favorable situation that $x_j$ is distributed according to the stationary distribution $(\mathfrak{s}_\pi, \pi)$, the *conditional* distribution of $x_{j+1}$ *given* $x_j$ can still be far from stationary, even though $x_{j+1}$ without conditioning is again stationary. The main trick used here is that $\eta$ is so small relative to the mixing time that $u_j$ evolves much more slowly than $x_j$, and thus any interaction between $x_j$ and $u_j$ can be replaced with an interaction between $x_j$ and $u_{j-k}$, which are *approximately* independent. The structure of the proof then is to first establish a few deterministic worst-case estimates on the behavior in any consecutive $k$ iterations, and then to perform an induction from $k$ to $N$.

For notational convenience, since $\pi$ is fixed in this proof, $\mathfrak{s}$ is written for $\mathfrak{s}_\pi$. Additionally, $\mathbb{E}_{\leq N}$ denotes the expectation over $((x_j, r_j))_{j\leq N}$, replacing the $\mathbb{E}_{\vec{x},\vec{r}}$ from the statement and allowing further flexibility by allowing the subscript to change.

**Worst case control between any $u_j$ and $u_{j-k}$.** Proceeding with this first part of the proof, define

$$\mathcal{T}_j(u) := x_j\left\langle x_j - \gamma x_{j+1}, u\right\rangle - x_j r_j,$$

whereby

$$u_{j+1} = u_j - \eta\mathcal{T}_j(u_j),$$
$$\|u_{j+1} - u_j\| = \eta\|x_j\left\langle x_j - \gamma x_j, u_j\right\rangle - x_j r_j\| \leq \eta\left((1+\gamma)\|u_j\| + 1\right),$$
$$\|u_j - u_{j-k}\| \leq \sum_{i=j-k}^{j-1}\|u_{i+1} - u_i\| \leq k\eta + \eta(1+\gamma)\sum_{i=j-k}^{j-1}\|u_j\|. \tag{C.1}$$

These inequalities will be useful in the induction as well, but now consider the first $k$ iterations.

**Controlling the first $k$ iterations.** Specifically, for any $i \leq k$, it will be established via induction that

$$\|u_i - \bar{u}\| \leq (1 + 1/(200k))^i\left(\frac{2}{1-\gamma}\right) + \eta\sum_{l<i}(1 + 1/(200k))^l\left(\frac{4}{1-\gamma}\right).$$

The base case follows since $u_0 = 0$ and $\|\bar{u}\| \leq 2/(1-\gamma)$. For the inductive step, since $\eta(1+\gamma) \leq 1/(200k)$,

$$\|u_{i+1} - \bar{u}\| = \|u_i - \bar{u} - \eta x_i\left\langle x_i - \gamma x_{i+1}, u_i - \bar{u} + \bar{u}\right\rangle + \eta x_i r_i\|$$
$$\leq (1 + \eta(1+\gamma))\|u_i - \bar{u}\| + \eta(1+\gamma)\|\bar{u}\| + \eta$$
$$\leq (1 + 1/(200k))^{i+1}\left(\frac{2}{1-\gamma}\right) + \eta\sum_{l<i}(1 + 1/(200k))^{l+1}\left(\frac{4}{1-\gamma}\right) + \eta\left(1 + \frac{2+2\gamma}{1-\gamma}\right)$$
$$\leq (1 + 1/(200k))^{i+1}\left(\frac{2}{1-\gamma}\right) + \eta\sum_{l<i+1}(1 + 1/(200k))^l\left(\frac{4}{1-\gamma}\right).$$

Since $(1 + 1/(200k))^i \le 2$ for all $i \le k$, then

$$\|u_i - \bar{u}\| \le \frac{4 + 8k\eta}{1 - \gamma} \le \frac{5}{1 - \gamma}. \tag{C.2}$$

This concludes the proof for the initial $k$ iterations.

**Controlling the remaining iterations via induction.** The rest of the proof now proceeds via induction on on iterations $k$ and higher: specifically, given $i \in \{k - 1, \dots, N - 1\}$, it will be shown that

$$\mathbb{E}_{\le N}\|u_{i+1} - \bar{u}\|^2 + \eta\mathbb{E}_{\le N} \sum_{j=k}^{i} \mathbb{E}_{x \sim (\mathfrak{s}, \pi)} \left\langle x, u_j - \bar{u} \right\rangle^2 \le \frac{26}{(1 - \gamma)^2}. \tag{C.3}$$

The base case $i = k - 1$ follows from eq. (C.2) (since the second term in the left hand side here is an empty sum), thus consider $i + 1$ with $i \ge k - 1$. The remainder of this inductive step will first introduce a variety of inequalities which need to hold for all $j \in \{k, \dots, i\}$, before returning to consideration of $i$ at the end.

Before continuing with the core argument for a fixed $j$, there are a few useful inequalities to establish, which will be used many times.

- Combining the inductive hypothesis with eq. (C.1),

$$\mathbb{E}\|u_j - u_{j-k}\|^2 \le 2k^2\eta^2 + 2\eta^2(1 + \gamma)^2 k \sum_{i=j-k}^{j-1} \mathbb{E}\|u_j - \bar{u} + \bar{u}\|^2.$$

$$\le 2k^2\eta^2 + 2\eta^2(1 + \gamma)^2 k \sum_{i=j-k}^{j-1} \left( \frac{56}{(1 - \gamma)^2} + \frac{2}{(1 - \gamma)^2} \right)$$

$$\le \frac{500k^2\eta^2}{(1 - \gamma)^2}. \tag{C.4}$$

  This is the explicit expression that will appear when moving between $u_j$ and $u_{j-k}$ to introduce (approximate) statistical independence.

- Next come a variety of bounds on $\mathcal{T}_j$. First, for any $j \le i$, using the inductive hypothesis and $\|\bar{u}\| \le 2/(1 - \gamma)$, for any $(x, x', r)$ and using the notation $\mathcal{T}_{x,x',r}(u) = x \left\langle x - \gamma x', u \right\rangle - xr$,

$$\mathbb{E}_{\le N}\|\mathcal{T}_{x,x',r}(u_j)\|^2 = \mathbb{E}_{\le N}\left\| x \left\langle x - \gamma x', u_j - \bar{u} + \bar{u} \right\rangle - xr \right\|^2$$

$$\le \mathbb{E}_{\le N} 4 \left( (1 + \gamma)^2\|u_j - \bar{u}\|^2 + (1 + \gamma)^2\|\bar{u}\|^2 + 1 \right) \tag{C.5}$$

$$\le \frac{4 \left( 26(1 + \gamma)^2 + 4(1 + \gamma)^2 + (1 - \gamma)^2 \right)}{(1 - \gamma)^2}$$

$$\le \frac{500}{(1 - \gamma)^2}. \tag{C.6}$$

  Separately, making use of eq. (C.4),

$$\mathbb{E}_{\le N}\|\mathcal{T}_j(u_j) - \mathcal{T}_j(u_{j-k})\|^2 = \mathbb{E}_{\le N}\left\| x_j \left\langle x_j - \gamma x_{j+1}, u_j - u_{j-k} \right\rangle \right\|^2$$

$$\le \mathbb{E}_{\le N}\|x_j\|^2 \cdot \|x_j - \gamma x_{j+1}\|^2 \cdot \|u_j - u_{j-k}\|^2$$

$$\le \frac{2000k^2\eta^2}{(1 - \gamma)^2}. \tag{C.7}$$

- Lastly, the convenience inequality which abstracts the application of mixing. Let $\mathbb{E}_{|j-k}$ denote the expectation of $(x_j, x_{j+1}, r_j)$ conditioned on all information up through time $j - k$, meaning $(x_l, r_l)_{l \le j-k}$. By the coupling characterization of total variation distance (Villani, 2008, Equation 6.11), there exists a joint distribution $\rho$ over two triples $(x_j, x_{j+1}, r_j)$ and $(x, x', r)$ where

the marginal distribution of $(x_j, x_{j+1}, r_j)$ is $\mathbb{E}_{|j-k}$, and the marginal distribution of $(z, z', s)$ is $(\mathfrak{s}, \pi)$, and crucially the choice of $k$ implies

$$\sup_{s_{j-k}} \Pr_\rho[(x_j, x_{j+1}, r_j) \neq (x, x', r)] \leq \sup_{s_{j-k}} \|P_\pi^k(s_{j-k}, \cdot) - \mathfrak{s}\|_{\mathrm{TV}} \leq m e^{-ck} \leq \frac{1}{N}.$$

(Note that $(x_j)_{j \leq N}$ inherit the mixing time for $(s_j)_{j \leq N}$ since $x_j = \mathrm{vec}(s_j a_j^\intercal)$, and $(a_j)_{j \leq N}$ are conditionally independent given $(s_j)_{j \leq N}$.) Using this inequality, and moreover making use of $\|\bar{u}\| \leq 2/(1-\gamma)$ and eqs. (C.5) and (C.6), and defining $\mathcal{T}_\mathfrak{s} := \mathbb{E}_{x,x',r \sim (\mathfrak{s},\pi)} \mathcal{T}_{x,x',r}$ to denote the update at stationarity,

$$\mathbb{E}_{\leq N} \left\langle u_{j-k} - \bar{u}, \mathcal{T}_\mathfrak{s}(u_{j-k}) - \mathcal{T}_j(u_{j-k}) \right\rangle$$

$$= \mathbb{E}_{\leq j-k} \left\langle u_{j-k} - \bar{u}, \mathcal{T}_\mathfrak{s}(u_{j-k}) - \mathbb{E}_{|j-k} \mathcal{T}_j(u_{j-k}) \right\rangle$$

$$= \mathbb{E}_{\leq j-k} \left\langle u_{j-k} - \bar{u}, \mathbb{E}_\rho \mathcal{T}_{x,x',r}(u_{j-k}) - \mathcal{T}_{x_j,x_{j+1},r_j}(u_{j-k}) \right\rangle$$

$$\leq \mathbb{E}_{\leq j-k} \|u_{j-k} - \bar{u}\| \left\| \mathbb{E}_\rho \mathcal{T}_{x_j,x_{j+1},r_j}(u_{j-k}) - \mathcal{T}_{x,x',r}(u_{j-k}) \right\|$$

$$\leq \sqrt{\mathbb{E}_{\leq j-k} \|u_{j-k} - \bar{u}\|^2} \sqrt{\mathbb{E}_{\leq j-k} \mathbb{E}_\rho \left\| \mathcal{T}_{x_j,x_{j+1},r_j}(u_{j-k}) - \mathcal{T}_{x,x',r}(u_{j-k}) \right\|^2}$$

$$\leq \sqrt{\frac{500 k^2 \eta^2}{(1-\gamma)^2}}$$

$$\cdot \sqrt{\mathbb{E}_{\leq j-k} \mathbb{E}_\rho \mathbb{1}[(x_j, x_{j+1}, r_j) \neq (x, x', r)] \cdot \left\| \mathcal{T}_{x_j,x_{j+1},r_j}(u_{j-k}) - \mathcal{T}_{x,x',r}(u_{j-k}) \right\|^2}$$

$$\leq \sqrt{\frac{5}{1600(1-\gamma)^2}}$$

$$\cdot \sqrt{\mathbb{E}_{\leq j-k} 16 \left(4\|u_{j-k} - \bar{u}\|^2 + 4\|\bar{u}\|^2 + 1\right) \mathbb{E}_\rho \mathbb{1}[(x_j, x_{j+1}, r_j) \neq (x, x', r)]}$$

$$\leq \sqrt{\frac{5}{1600(1-\gamma)^2}} \sqrt{\frac{1}{N} \mathbb{E}_{\leq j-k} 16 \left(4\|u_{j-k} - \bar{u}\|^2 + 4\|\bar{u}\|^2 + 1\right)}$$

$$\leq \sqrt{\frac{5}{1600(1-\gamma)^2}} \sqrt{\frac{2000}{N(1-\gamma)^2}}$$

$$\leq \frac{3}{(1-\gamma)^2 \sqrt{N}}. \tag{C.8}$$

Now comes the main part of the inductive step. Expanding the square and making one appeal to eq. (C.6),

$$\mathbb{E}_{\leq N} \|u_{j+1} - \bar{u}\|^2 = \mathbb{E}_{\leq N} \|u_j - \bar{u}\|^2 - 2\eta \mathbb{E}_{\leq N} \left\langle u_j - \bar{u}, \mathcal{T}_j(u_j) \right\rangle + \eta^2 \mathbb{E}_{\leq N} \|\mathcal{T}_j(u_j)\|^2$$

$$\leq \mathbb{E}_{\leq N} \|u_j - \bar{u}\|^2 - 2\eta \mathbb{E} \left\langle u_j - \bar{u}, \mathcal{T}_j(u_j) \right\rangle + \frac{500\eta^2}{(1-\gamma)^2}. \tag{C.9}$$

To lower bound the middle term, making extensive use of eqs. (C.4), (C.6) and (C.7) combined with the Cauchy-Schwarz inequality, and the inductive hypothesis to control $\mathbb{E}_{\leq N} \|u_{j-k} - \bar{u}\|^2$,

$$\mathbb{E}_{\leq N} \left\langle u_j - \bar{u}, \mathcal{T}_j(u_j) \right\rangle \geq \mathbb{E}_{\leq N} \left\langle u_{j-k} - \bar{u}, \mathcal{T}_j(u_j) \right\rangle - \mathbb{E}_{\leq N} \|u_{j-k} - u_j\| \cdot \|\mathcal{T}_j(u_j)\|$$

$$\geq \mathbb{E}_{\leq N} \left\langle u_{j-k} - \bar{u}, \mathcal{T}_j(u_{j-k}) \right\rangle - \mathbb{E}_{\leq N} \|u_{j-k} - u_j\| \cdot \|\mathcal{T}_j(u_j)\|$$

$$\quad - \mathbb{E}_{\leq N} \|u_{j-k} - \bar{u}\| \cdot \|\mathcal{T}_j(u_j) - \mathcal{T}_j(u_{j-k})\|$$

$$\geq \mathbb{E}_{\leq N} \left\langle u_{j-k} - \bar{u}, \mathcal{T}_j(u_{j-k}) \right\rangle - \sqrt{\mathbb{E}_{\leq N} \|u_{j-k} - u_j\|^2} \sqrt{\mathbb{E}_{\leq N} \|\mathcal{T}_j(u_j)\|^2}$$

$$\quad - \sqrt{\mathbb{E}_{\leq N} \|u_{j-k} - \bar{u}\|^2} \sqrt{\mathbb{E}_{\leq N} \|\mathcal{T}_j(u_j) - \mathcal{T}_j(u_{j-k})\|^2}$$

$$\geq \mathbb{E}_{\leq N} \left\langle u_{j-k} - \bar{u}, \mathcal{T}_j(u_{j-k}) \right\rangle - \frac{500 k \eta}{(1-\gamma)^2} - \frac{250 k \eta}{(1-\gamma)^2}$$

$$\geq \mathbb{E}_{\leq N} \left\langle u_{j-k} - \bar{u}, \mathcal{T}_j(u_{j-k}) \right\rangle - \frac{750 k \eta}{(1-\gamma)^2}.$$

Now comes the key step of the proof, which uses the mixing time and introduced gap of size $k$ to replace $\mathcal{T}_j(u_{j-k})$ with $\mathcal{T}_{\mathfrak{s}}(u_{j-k})$; this reasoning is captured in eq. (C.8), which gives

$$\mathbb{E}_{\leq N}\left\langle u_{j-k} - \bar{u}, \mathcal{T}_j(u_{j-k})\right\rangle = \mathbb{E}_{\leq N}\left\langle u_{j-k} - \bar{u}, \mathcal{T}_{\mathfrak{s}}(u_{j-k})\right\rangle - \mathbb{E}_{\leq N}\left\langle u_{j-k} - \bar{u}, \mathcal{T}_{\mathfrak{s}}(u_{j-k}) - \mathcal{T}_j(u_{j-k})\right\rangle$$
$$\geq \mathbb{E}_{\leq N}\left\langle u_{j-k} - \bar{u}, \mathcal{T}_{\mathfrak{s}}(u_{j-k})\right\rangle - \frac{3}{(1-\gamma)^2\sqrt{N}}.$$

Again continuing with the non-constant term, and using the general inequality $2ab \leq a^2 + b^2$ which holds for any reals $a, b$, letting $x \sim (\mathfrak{s}, \pi)$ and $x' \sim x$ and $r$ a random reward (i.e., all the random quantities used to construct $\mathcal{T}_{\mathfrak{s}}$, though $r$ will cancel), and lastly introducing $\mathcal{T}_{\mathfrak{s}}(\bar{u})$ via the fixed point property in the form $\mathcal{T}_{\mathfrak{s}}(\bar{u}) = 0$, and the fact that $x'$ has the same distribution as $x$ when it appears alone,

$$\left\langle u_{j-k} - \bar{u}, \mathcal{T}_{\mathfrak{s}}(u_{j-k})\right\rangle = \left\langle u_{j-k} - \bar{u}, \mathcal{T}_{\mathfrak{s}}(u_{j-k}) - \mathcal{T}_{\mathfrak{s}}(\bar{u})\right\rangle$$
$$= \mathbb{E}_{x,r,x'}\left\langle x, u_{j-k} - \bar{u}\right\rangle\left\langle x - \gamma x', u_{j-k} - \bar{u}\right\rangle$$
$$= \mathbb{E}_{x,r,x'}\left\langle x, u_{j-k} - \bar{u}\right\rangle^2 - \gamma\left\langle x, u_{j-k} - \bar{u}\right\rangle\left\langle x', u_{j-k} - \bar{u}\right\rangle$$
$$\geq \mathbb{E}_{x,r,x'}\left\langle x, u_{j-k} - \bar{u}\right\rangle^2 - \gamma/2\left(\left\langle x, u_{j-k} - \bar{u}\right\rangle^2 + \left\langle x, u_{j-k} - \bar{u}\right\rangle^2\right)$$
$$= (1-\gamma)\mathbb{E}_x\left\langle x, u_{j-k} - \bar{u}\right\rangle^2,$$

where

$$\mathbb{E}_{\leq N}\mathbb{E}_x\left\langle x, u_{j-k} - \bar{u}\right\rangle^2 = \mathbb{E}_{\leq N}\mathbb{E}_x\left\langle x, u_j - \bar{u}\right\rangle^2 + 2\mathbb{E}_{\leq N}\mathbb{E}_x\left\langle x, u_j - \bar{u}\right\rangle\left\langle x, u_{j-k} - u_j\right\rangle$$
$$+ \mathbb{E}_{\leq N}\mathbb{E}_x\left\langle x, u_{j-k} - u_j\right\rangle^2$$
$$\geq \mathbb{E}_{\leq N}\mathbb{E}_x\left\langle x, u_j - \bar{u}\right\rangle^2 - 2\sqrt{\mathbb{E}_{\leq N}\|u_j - \bar{u}\|^2}\sqrt{\mathbb{E}_{\leq N}\|u_{j-k} - u_j\|^2}$$
$$\geq \mathbb{E}_{\leq N}\mathbb{E}_x\left\langle x, u_j - \bar{u}\right\rangle^2 - \frac{2\sqrt{26 \cdot 500k^2\eta^2}}{(1-\gamma)^2}$$
$$\geq \mathbb{E}_{\leq N}\mathbb{E}_x\left\langle x, u_j - \bar{u}\right\rangle^2 - \frac{800k\eta}{(1-\gamma)^2}.$$

Plugging all of this back in to eq. (C.9) gives

$$\mathbb{E}_{\leq N}\|u_{j+1} - \bar{u}\|^2 \leq \mathbb{E}_{\leq N}\|u_j - \bar{u}\|^2 - \eta(1-\gamma)\mathbb{E}_{\leq N}\mathbb{E}_x\left\langle x, u_j - \bar{u}\right\rangle^2 + \frac{1600k\eta^2 + 3\eta/\sqrt{N}}{(1-\gamma)^2},$$

which after summing over all $j \in \{k, \ldots, i\}$ and telescoping and re-arranging gives

$$\mathbb{E}_{\leq N}\|u_{i+1} - \bar{u}\|^2 + \sum_{j=k}^{i}\eta(1-\gamma)\mathbb{E}_{\leq N}\mathbb{E}_x\left\langle x, u_j - \bar{u}\right\rangle^2 \leq \mathbb{E}_{\leq N}\|u_k - \bar{u}\|^2 + \sum_{j=k}^{i}\frac{1600k\eta^2 + 3\eta/\sqrt{N}}{(1-\gamma)^2}$$
$$\leq \frac{25}{(1-\gamma)^2} + N\left(\frac{1600k\eta^2 + 3\eta/\sqrt{N}}{(1-\gamma)^2}\right)$$
$$\leq \frac{26}{(1-\gamma)^2},$$

which establishes the inductive hypothesis stated in eq. (C.3).

**Final cleanup.** To finish the proof, a tiny amount of cleanup is needed. Adding the missing prefix of the sum from the conclusion of the induction gives

$$\mathbb{E}_{\leq N}\|u_N - \bar{u}\|^2 + \sum_{j < N}\eta(1-\gamma)\mathbb{E}_{\leq N}\mathbb{E}_x\left\langle x, u_j - \bar{u}\right\rangle^2 \leq \frac{26}{(1-\gamma)^2} + \sum_{j < k}\eta(1-\gamma)\mathbb{E}_{\leq N}\mathbb{E}_x\left\langle x, u_j - \bar{u}\right\rangle^2$$
$$\leq \frac{26}{(1-\gamma)^2} + \frac{25k\eta(1-\gamma)}{(1-\gamma)^2}$$
$$\leq \frac{27}{(1-\gamma)^2}.$$

The final bound now follows by using Jensen's inequality to introduce $\hat{u}_N$ in the summation term, and introducing $\hat{u}_N$ within the norm term by noting the bound held for all $i < N$ and thus the triangle inequality implies $\|\hat{u} - \bar{u}\| \leq \sum_{i<N} \|u_i - \bar{u}\|/N \leq \sqrt{27}/(1 - \gamma)$, whose square can be added to both sides to give the final bound. $\qquad\square$

These proofs immediately imply Lemma 3.3.

*Proof of Lemma 3.3.* Lemma 3.3 is a restatement of Lemma C.4, but using a combination of Assumption 1.3 and Lemma C.3 (and in particular the fixed point $\bar{u}$ defined in the latter) to simplify Lemma C.4. $\qquad\square$

## D  PROOF OF THEOREM 1.4

Combining the mirror descent and sampling tools, we can finally prove Theorem 1.4.

*Proof of Theorem 1.4.* Throughout this proof, let $\delta > 0$ denote a unit of failure probability; the final bound will use the choice $\delta := t^{-9/8}/2$, although most of the proof will simply write $\delta$ for sake of interpretation.

Define the following KL-bounded subset of policy space:

$$\mathcal{P} := \bigcap_{s \in \mathcal{S}} \mathcal{P}_s, \qquad \text{where } \mathcal{P}_s := \left\{ \pi : K_{\mathfrak{v}_{\bar{\pi}}^s}(\overline{\pi}, \pi) \leq \ln k + \frac{1}{(1 - \gamma)^2} \right\}.$$

By $|\mathcal{S}|$ applications of Lemma 3.1 (one for each $\mathcal{P}_s$) and taking maxima/minima of the resulting constants, since $\mathfrak{s}_{\bar{\pi}}$ is positive for every state, then there exist constants $p_{\min} > 0$, $C_1 > 0$, and $C_2 \geq 1$ so that, for any $\pi \in \mathcal{P}$ and any state $s$ and any optimal action $a \in \mathcal{A}_s$,

$$p_{\min} \leq \mathfrak{s}_{\pi}(s), \qquad \pi(s, a) \geq \frac{\overline{\pi}(s, a)}{C_2},$$

and letting $P_\pi$ denote the transition matrix on the induced chain on $\mathcal{S}$, for any time $q$,

$$\max_{s \in \mathcal{S}} \|P_\pi^q(s) - \mathfrak{s}_\pi\|_{\mathrm{TV}} \leq C_2 \exp(-C_1 q). \tag{D.1}$$

Lastly, the fully specified parameters $N$ and $\eta$ are

$$N := \frac{10^7 t^2 C_2^4 \ln C_2}{p_{\min}^4 C_1} \ln \left( \frac{10^7 t^2 C_2^4 \ln C_2}{p_{\min}^4 C_1} \right), \quad \eta := \frac{1}{400\sqrt{kN}}, \quad \text{where } k := \left\lceil \frac{\ln N + \ln C_2}{C_1} \right\rceil,$$

which after expanding the choice of $\theta$ satisfy

$$N = \Theta\left( t^2 \ln t \right), \qquad \eta = \Theta\left( \frac{1}{\sqrt{N \ln N}} \right), \qquad \frac{1}{N\eta} \leq \frac{p_{\min}^2}{4tC_2^2},$$

where the first two match the desired statement, and the last inequality is used below.

The proof establishes the following inequalities inductively: defining $\varepsilon_j$ for convenience as

$$\varepsilon_j := \sup_{s,a} \left( \widehat{\mathcal{Q}}_i(s, a) - \mathcal{Q}_i(s, a) \right)^2 + \eta N \mathbb{E}_{(s,a)\sim(\mathfrak{s}_\pi, \pi)} \left( \widehat{\mathcal{Q}}_i(s, a) - \mathcal{Q}_i(s, a) \right)^2,$$

then with probability at least $1 - 2i\delta$,

$$K_{\mathfrak{v}_{\bar{\pi}}^s}(\overline{\pi}, \pi_i) + \theta(1 - \gamma) \sum_{j<i} \left( \mathcal{V}_j(s) - \mathcal{V}_{\overline{\pi}}(s) \right) \quad \leq \quad \ln k + \frac{1}{(1 - \gamma)^2}, \qquad \forall s, \qquad \text{(IH.MD)}$$

$$\mathbb{E}_{\widehat{\mathcal{Q}}_j} \varepsilon_j \quad \leq \quad \frac{54}{(1 - \gamma)^2}, \qquad \forall j \leq i, \quad \text{(IH.TD.1)}$$

$$\varepsilon_j \quad \leq \quad \frac{54}{\delta(1 - \gamma)^2}, \qquad \forall j \leq i, \quad \text{(IH.TD.2)}$$

where $\mathbb{E}_{\widehat{\mathcal{Q}}_j}$ denotes the expectation over the new $N$ examples used to construct $\widehat{\mathcal{Q}}_j$ but conditions on the prior samples. The first inequality, eq. (IH.MD), implies $\pi_i \in \mathcal{P}$ directly, and moreover implies the final statement when $i = t$ after plugging in $\delta = t^{-9/8}/2$ and rearranging.

To establish the inductive claim, consider some $i > 0$ (the base case $i = 0$ comes for free), and suppose the inductive hypothesis holds for $i - 1$; namely, discard its $2(i-1)\delta$ failure probability, and suppose the three inequalities hold. This induction will first handle the mirror descent guarantee in eq. (IH.MD), and then establish the TD guarantees in eqs. (IH.TD.1) and (IH.TD.2) together.

The first inequality, eq. (IH.MD), is established via the simplified mirror descent bound in Lemma 2.1. To start, since $K_\nu(\overline{\pi}, \pi_0) \le \ln k$ for any measure $\nu$, then instantiating the simplified bound in Lemma 2.1 for the first $i$ iterations for any starting state $s$ gives

$$
\begin{aligned}
& K_{\mathfrak{v}_{\overline{\pi}}^s}(\overline{\pi}, \pi_i) + \theta(1-\gamma) \sum_{j<i} \left( \mathcal{V}_{\overline{\pi}}(s) - \mathcal{V}_j(s) \right) \\
& \le \quad \ln k + \theta \sum_{j<i} \left\langle \mathcal{Q}_j - \widehat{\mathcal{Q}}_j, \pi_j - \overline{\pi} \right\rangle_{\mathfrak{v}_{\overline{\pi}}^s} + \theta^2 \sum_{j<i} \sup_{s,a} \widehat{\mathcal{Q}}_j(s,a)^2.
\end{aligned}
\tag{D.2}
$$

Upper bounding the second two terms will make use of upper bounds on $(\varepsilon_j)_{j<i}$, but rather than using only eq. (IH.TD.2), here is a more refined approach. For each $j < i$, define an indicator random variable

$$
F_j := \mathbb{1}\left[ \varepsilon_j > \frac{54\sqrt{t}}{(1-\gamma)^2} \right],
$$

which by eq. (IH.TD.1) and Markov's inequality (since $\varepsilon_j \ge 0$) satisfies

$$
\Pr_{\widehat{\mathcal{Q}}_j}(F_j) \le \Pr_{\widehat{\mathcal{Q}}_j}\left[ \varepsilon_j > \sqrt{t}\mathbb{E}_{\widehat{\mathcal{Q}}_j} \varepsilon_j \right] \le \frac{1}{\sqrt{t}}.
$$

By Azuma's inequality applied to the $i$ binary random variables $(F_j)_{j<i}$ (where $F_j - \mathbb{E}_{\widehat{\mathcal{Q}}_j} F_j$ forms a Martingale difference sequence since $\mathbb{E}_{\widehat{\mathcal{Q}}_j}$ conditions on the old sequence and takes the expectation over the $N$ new samples), with probability at least $1 - \delta$, and plugging in the choice $\delta = t^{-9/8}/2$,

$$
\sum_{j<i} F_j \le \sum_{j<i} \Pr(F_j) + \sqrt{\frac{i}{2} \ln \frac{1}{\delta}} \le \sqrt{i}\left( 1 + \sqrt{\frac{9}{16} \ln 2t} \right) \le \sqrt{3t \ln t};
$$

henceforth discard this failure probability, bringing the total failure probability to $(2i-1)\delta$. Combining this with eq. (IH.TD.2) gives

$$
\begin{aligned}
\sum_{j<i} \varepsilon_j &\le \sum_{\substack{j<i \\ F_j=1}} \varepsilon_j + \sum_{\substack{j<i \\ F_j=0}} \varepsilon_j \\
&\le \sum_{\substack{j<i \\ F_j=1}} \frac{54}{\delta(1-\gamma)^2} + \sum_{\substack{j<i \\ F_j=0}} \frac{54\sqrt{t}}{(1-\gamma)^2} \\
&\le \frac{54}{(1-\gamma)^2} \left( \frac{\sqrt{3t \ln t}}{\delta} + t^{3/2} \right) \\
&\le \frac{270 t^{13/8}\sqrt{\ln t}}{(1-\gamma)^2} \le \frac{1}{8\theta^2(1-\gamma)^2}.
\end{aligned}
$$

Turning back to eq. (D.2), this gives a way to control the middle term: since $\overline{\pi}(s, b) = 0$ for $b \notin \mathcal{A}_s$,

$$\sum_{j<i} \left\langle \mathcal{Q}_j - \widehat{\mathcal{Q}}_j, \pi_j - \overline{\pi} \right\rangle_{\mathfrak{v}^s_{\overline{\pi}}} \le \sum_{j<i} \max_s \left\langle \mathcal{Q}_j - \widehat{\mathcal{Q}}_j, \pi_j - \overline{\pi} \right\rangle_{\delta_s}$$

$$\le \frac{1}{p_{\min}} \sum_{j<i} \left\langle \mathcal{Q}_j - \widehat{\mathcal{Q}}_j, \pi_j - \overline{\pi} \right\rangle_{\mathfrak{s}_{\pi_j}}$$

$$= \frac{1}{p_{\min}} \sum_{j<i} \left( \underset{(s,a)\sim(\mathfrak{s}_j, \pi_j)}{\mathbb{E}} (\mathcal{Q}_j(s,a) - \widehat{\mathcal{Q}}_j(s,a)) \right.$$

$$\left. + \underset{s\sim\mathfrak{s}_j}{\mathbb{E}} \sum_{a\in\mathcal{A}_s} \overline{\pi}(s,a)(\widehat{\mathcal{Q}}_j(s,a) - \mathcal{Q}_j(s,a)) \right)$$

$$\le \frac{2C_2}{p_{\min}} \sum_{j<i} \underset{(s,a)\sim(\mathfrak{s}_j, \pi_j)}{\mathbb{E}} |\mathcal{Q}_j(s,a) - \widehat{\mathcal{Q}}_j(s,a)|$$

$$\le \frac{2C_2\sqrt{i}}{p_{\min}} \sqrt{\sum_{j<i} \underset{(s,a)\sim(\mathfrak{s}_j, \pi_j)}{\mathbb{E}} (\mathcal{Q}_j(s,a) - \widehat{\mathcal{Q}}_j(s,a))^2}$$

$$\le \frac{2C_2\sqrt{i}}{p_{\min}\sqrt{N\eta}} \sqrt{\sum_{j<i} \varepsilon_j}$$

$$\le \frac{1}{2\theta(1-\gamma)}.$$

Meanwhile, for the last term in eq. (D.2), since $\sup_{s,a} \mathcal{Q}_j(s,a) \le 1/(1-\gamma)$,

$$\sum_{j<i} \sup_{s,a} \widehat{\mathcal{Q}}_j(s,a)^2 \le 2 \sum_{j<i} \sup_{s,a} \left[ \mathcal{Q}_j(s,a)^2 + \left( \widehat{\mathcal{Q}}_j(s,a) - \mathcal{Q}_j(s,a) \right)^2 \right]$$

$$\le \frac{2t}{(1-\gamma)^2} + 2\sum_{j<i} \epsilon_j$$

$$\le \frac{1}{2\theta^2(1-\gamma)^2}.$$

Plugging all of this back in to eq. (D.2) gives

$$K_{\mathfrak{v}^s_{\overline{\pi}}}(\overline{\pi}, \pi_i) + \theta(1-\gamma) \sum_{j<i} \left( \mathcal{V}_{\overline{\pi}}(s) - \mathcal{V}_j(s) \right)$$

$$\le \quad \ln k + \theta \sum_{j<i} \left\langle \mathcal{Q}_j - \widehat{\mathcal{Q}}_j, \pi_j - \overline{\pi} \right\rangle_{\mathfrak{v}^s_{\overline{\pi}}} + \theta^2 \sum_{j<i} \sup_{s,a} \widehat{\mathcal{Q}}_j(s,a)^2$$

$$\le \quad \ln k + \frac{1}{(1-\gamma)^2},$$

thus concluding the proof of eq. (IH.MD) and the mirror descent part of the inductive step.

The TD part of the inductive step is now direct: by eq. (IH.MD), then $\pi_i \in \mathcal{P}$, and thus eq. (IH.TD.1) follows directly from Lemma 3.3, and eq. (IH.TD.2) follows via Markov's inequality after discarding another $\delta$ failure probability, bringing the total failure probability to $2i\delta$, and completing the inductive step and overall proof. $\qquad\square$

