# OpenReview forum: "Actor-critic is implicitly biased towards high entropy optimal policies"
_ICLR.cc/2022/Conference — ICLR 2022 Poster_

### Official Review · Reviewer_gsmP · 2021-10-26

**Correctness:** 4
**Technical Novelty And Significance:** 4
**Empirical Novelty And Significance:** 4
**Recommendation:** 8
**Confidence:** 4

**Main Review:**

I think the paper makes a reasonable contribution to the field.
First, it explicitly uses the results from mirror descent to analyze their actor update, which confirms that once the policy is within the ball of the maximum entropy optimal policy, it remains there forever provided that the critic is sufficiently accurate. This result appears novel to my knowledge and is useful in eliminating previously used strong assumptions.
Second, it analyzes the converge of linear TD without using a projection, which also appears novel to my knowledge.

However, I do have some concerns. I find the proofs are not very reader-friendly. I cannot really verify the proof of Lemma C.2, though I tend to believe it is correct. I think the coupling used in page 18 is not clearly defined. What is the distribution of z_{i, j} and can E_{i, j} be clearly defined? And I cannot get why the inequality leading to 32m / B ... holds. This is the major concern for my score 5; I'd like to raise my score if  this concern is clarified. In general, I think there are many skipped steps in the proofs, it would be good if all the proofs can be verified directly by looking at the pdf without doing extra computation.

A possible improvement: I am not sure if linear MDP is a necessary ingredient for this work. In my understanding, it's used only to ensure that the TD fixed point is the true value function. I think this can also be ensured by either considering tabular setting or linear function approximation with compatible features. It would be good if the authors can briefly discuss this, after all linear MDP is a very strong assumption.


**Summary Of The Paper:**

The paper proposes and analyzes the convergence rate and optimality of a natural-actor-critic-like actor critic with linear function approximation in linear MDPs. The most important discovery is that once the policy is within the ball of the maximum entropy optimal policy, it remains there forever provided that the critic is sufficiently accurate. Consequently, one now needs to make ergodicity assumption for only the optimal policy, instead of all policies along the optimization path.

**Summary Of The Review:**

I think the paper makes reasonable contribution but the presentation of the proofs can be improved.

==============

The authors successfully addressed my concerns and I increased my score accordingly.

---

> ### Author Response · Authors · 2021-11-18
> **Thank you for your review.  (Response part 1 of 2.)**
>
> We thank the reviewer for their comments, time, and support.  We have heavily revised the
> submission in response to their detailed comments, which we greatly appreciate.
> Our response is long, so here is a brief summary:
>
> - We agree that our original TD proof was too short and hard to verify; the new one is
>   roughly twice as long, explicit, and better organized.  We removed the mini-batching
>   to improve readability and allow direct comparison to prior work,
>   but the proof ideas are the same.
>
> - We agree that compatible linear function approximation suffices, and have updated our TD
>   analysis and related work discussion to reflect this.
>
> As follows are our detailed responses, split across two openreview comments:
>
> - **"I cannot get why the inequality leading to 32m / B ... holds".**
>   We apologize for the original abbreviated proof.
>   We have greatly expanded and re-organized the proof, and
>   will summarize its key points below.
>   In this process, we realized the same proof idea goes through without mini-batching,
>   and went ahead with this disruptive change because it improves presentation
>   (no triple subscripts!)
>   and allows direct comparison to prior TD analyses.
>   We can now say that, for the exact same TD algorithm, prior work needs not only
>   projections *but also* to start from stationarity (or even stronger probabilistic assumptions).
>
>   Returning to the specific "32m / B ..." step, we can say we use three ideas overall,
>   which are explicit in the new version of the proof,
>   and summarized as follows:
>     - The inductive hypothesis gives us a control on $\\|u_j - \bar u\\|$, which is used throughout;
>       this is due to an implicit bias, and is essentially the same in the new and old proofs.
>     - The main technical issue is the statistical dependence between $x_j$ and $u_j$.
>       Intuitively, this is fine if the chain mixes fast and step sizes are small, but here is
>       more detail.  In the new proof, whenever we run into this issue, we add and subtract
>       a $u_{j-k}$ for some appropriately chosen $k$, and instead consider the action
>       of $u_{j-k}$ on $x_j$.  Due to fast mixing, these two are nearly independent, whereas due
>       to small step sizes, $u_j - u_{j-k}$ is small.  In the old proof, this same idea was used,
>       but occurred within mini-batches, and was not written out in sufficient detail,
>       and we feel is the main source of concern for the "32m / B ..." step.
>     - (This point also relates to **"what is the distribution of z_{i, j} and can E_{i, j} be clearly defined?"**.)  Our $x_j$ is never completely mixed, so we consider its relationship
>       to a $y_j$ which is distributed exactly according the stationary distribution.
>       Indeed, we can view the distribution of $x_j$ as a mixture of two distributions:
>       one is $y_j$, and we simply call the other $z_j$, which for now we only know is bounded
>       since $x_j$ is bounded.
>       The event $E_j$ corresponds to the mixture $x_j$ choosing to sample from component $y_j$,
>       and analogously $\lnot E_j$ means we sample from $z_j$;
>       in this way, we obtain an implicit definition of $z_j$ (as the "remaining" mass of $x_j$).
>       We do not construct an explicit coupling ourselves,
>       rather the existence of these coupled random variables is provided
>       by a coupling characterization of total variation distance,
>       which we use to control the probability of $\lnot E_j$:
>       specifically, $\text{Pr}[\lnot E_j] = \text{TV}(x_j,\text{stationary})$
>       (see appropriate references in the new version),
>       and now we can
>       invoke mixing time tools to upper bound this.
>       Lastly, we can not take credit for this idea,
>       a version of it appears in the literature on volume estimation via random walks,
>       and we have now included a proper citation.
>   We thank the reviewer again for their detailed comments,
>   and look forward to further discussion!
>
> (Our response continues below in a second openreview comment; thank you for your time and patience.)

---

> > ### Author Response · Authors · 2021-11-18
> > **Thank you for your review. (Response part 2 of 2.)**
> >
> > (Continuation of previous response.)
> >
> > - **"I think this can also be ensured by either considering tabular setting or
> >   linear function approximation with compatible features."** We thank the reviewer
> >   for pointing this out, and confirm that
> >   both the tabular setting and the compatible linear function approximation
> >   settings work, and have revised the submission as follows.
> >   Firstly, in the related work, we have included two references for compatible
> >   linear function approximation, and discussed both how they can be used to relate
> >   our algorithm to others in the literature, and that our analysis goes through if
> >   the compatible assumption is made *for every policy $\pi_i$ and corresponding $\mathcal Q_i$
> >   encountered within the algorithm* (we will return to this point shortly).
> >   Secondly, when we removed mini-batching from our core TD analysis (Lemma C.4),
> >   we re-wrote it
> >   so that we do not invoke any of our MDP assumptions (e.g., finite state space or linearity);
> >   Lemma C.4 is now stated in generality for TD fixed points,
> >   and both the compatible approximation assumption or the linear MDP assumption can be
> >   invoked to treat the fixed point as the true Q function.
> >
> >   That said, as we noted, the compatibility assumption
> >   would need to be made for every $\mathcal Q_i$
> >   considered throughout the algorithm.  Since $\pi_i$ is a random variable, it is hard for
> >   us to interpret such an assumption, and as a matter of taste, we prefer the "global" assumption
> >   of a linear MDP, even though we agree it is a stronger assumption.
> >   How does the reviewer feel about this perspective?
> >
> >   We thank the reviewer for this comment, and look forward to further discussion!
> >
> > - **"Second, it analyzes the converge of linear TD without using a projection, which also appears novel to my knowledge."**  Now that we removed the use of mini-batches in our TD, we can directly
> >   compare to prior work, and in addition to dropping projection,
> >   we can also say that prior work could not start from arbitrary states as we do, but needed
> >   to be started from the stationary distribution of the policy.
> >   We have clarified this point in
> >   our revisions, and hope it is a valuable contribution to the TD literature.
> >
> > - **"Once the policy is within the ball of the maximum entropy optimal policy,
> >   it remains there forever".**  A quick clarification, our proof handles all
> >   iterations, including the initial ones, and does not require any assumption
> >   that we reach some small KL ball.  It is true that we stated our results for a specific
> >   choice of initial policy (namely, using $W_0 = 0$), but our analysis goes through for
> >   any initial policy: concretely, notice that our mirror descent bound (Lemma 2.1) has a KL divergence
> >   against the initial policy on the right hand side, and we could leave this term in, rather
> >   than upper bounding it with $\ln k$ which comes as a consequence of $W_0 = 0$.  If the
> >   reviewer has further comments or would like us to expanding the relevant discussion in the
> >   submission, we will gladly do so.
> >
> > - **"It would be good if all the proofs can be verified directly by looking at the pdf without doing extra computation".**  We have greatly expanded the proofs in the appendix, and also added sketches for all proofs in the body.  We agree this is important and apologize for the difficulties in our original submission; we look forward to further feedback from the reviewer!
> >
> >
> > We thank the reviewer once again, and look forward to further comments, and to the resulting improvements to our submission!

---

> > > ### Comment · Reviewer_gsmP · 2021-11-24
> > > **Concern about the equality above C.5**
> > >
> > > Thanks for expanding the proofs. I'm now trying to check the TD proofs and have some concerns about the equation above C.5, in particular, Pr[-E_j] = ||s_j - s||_TV. In my understanding, the random variable pair (x_j, y_j) is a coupling of (s_j, s). So according to Prop 7 of Levin et al., 2006, we have Pr[-E_j] = Pr(x_j != y_j) \geq ||s_j - s||_{TV}. But why do we have equality in the paper? Indeed, as stated in Remark 8 of Levin et al., 2006, there are random variable pairs that achieve the equality. But that equality-achieving (optimal coupling) random variables are not necessarily identical to the (x_j, y_j) you defined. Have I missed something obvious here?

---

> > > > ### Author Response · Authors · 2021-11-25
> > > > **Thank you for your patience!**
> > > >
> > > >
> > > > Dear reviewer gsmP,
> > > >
> > > > Thank you very much for your patience with us; we realize now that this is
> > > > still the same concern you had before, and we apologize for not addressing it
> > > > satisfactorily.  We believe we understand the concern now and have
> > > > a satisfactory answer.
> > > >
> > > > We agree that our invocation of Levin et al's Proposition 4.7 is as though the random variables we
> > > > sketched exactly satisfy the equality, but not only do we not give any indication
> > > > why this should be true, but moreover, as we have not crisply defined the
> > > > random variables, it is unreasonable to expect a reader to verify the equality.
> > > >
> > > > As follows is our explanation, and separately how we plan to fix it.
> > > >
> > > > - **(Explanation.)** The exact coupling we use is in fact in the *proof* of Proposition 4.7;
> > > >   for some reason, we had misremembered and believed the construction is stated as part of
> > > >   the proposition statement.
> > > >   In detail, if we can trouble you to open the book to Proposition 4.7 once
> > > >   again, notice within its proof, just after eq. (4.13),
> > > >   the discussion "flip a coin with probability of heads
> > > >   equal to $p$".  This coin precisely corresponds to our $E_j$: when it lands heads,
> > > >   the discussion just after eq. (4.13), starting from the bottom of page 51, establishes that
> > > >   $x_j = y_j$ in our notation, and following the rest of the proof leads to an explicit
> > > >   construction of our $E_j$, $y_j$, and $z_j$ in general (see [*] below for detail).
> > > >   We believe this is exactly the clarification you were seeking, both now and in your
> > > >   original review; is that the case?
> > > >
> > > >   We apologize again that this construction is not provided as part of the
> > > >   statement of Proposition 4.7 or as part of our text.
> > > >   In fact, part of why we were sloppy with this
> > > >   is that in the volume estimation literature we mention, namely the survey by
> > > >   Santosh Vempala (2005) which we cite, this construction is treated
> > > >   as a "folklore" fact and given without citation or proof: to see what we
> > > >   mean, if you have the survey open, search for the text "divine intervention",
> > > >   which gives a succinct description of the construction and its properties
> > > >   with no proof or citation
> > > >   (should be page 28/40 of the PDF, labeled as page 600).
> > > >   We also found this fact presented similarly without citation or proof
> > > >   in works which this survey cites.
> > > >
> > > >   [*]  A precise correspondence will have confusing notation, because they use a single pair
> > > >   $(X,Y)$, whereas we use a triple $(x_j,y_j,z_j)$ and care about different parts under
> > > >   $E_j$ and under $\lnot E_j$.  In detail, we can take their $Y$ to correspond to our $y_j$,
> > > >   which is distributed according to the stationary distribution.
> > > >   Under event $E_j$ (their "heads"), we have $X=x_j = y_j = Y$ (they prove $X=Y$).
> > > >   On the other hand, when $\lnot E_j$ occurs (their "tails"), they establish $X \neq Y$,
> > > >   and thus still taking $Y=y_j$, now $X= x_j = z_j \neq y_j$.  This construction is adequate for our proof:
> > > >   we get a term with $E_j$, where we replace $x_j$ with $y_j$ whereby a Bellman operator
> > > >   becomes one at stationarity, and in the term with $\lnot E_j$, we only need to bound $\Pr[\lnot E_j]$,
> > > >   and do not need the structure of $z_j$ in more detail.
> > > >
> > > > - **(Revision plan.)**  We agree that we need to crisply define these random variables, and it
> > > >   is not adequate to say "see the proof of Proposition 4.7 in Levin et al." or
> > > >   "see a discussion of this property (without proof) in Vempala (2005)".
> > > >   Therefore our revisions will give an explicit construction (as in Levin et al's proof),
> > > >   and then either cite or prove its correctness.
> > > >
> > > > Does this sound satisfactory?
> > > >
> > > > Thank you once again for your careful reading; we look forward to further discussion!

---

> > > > > ### Comment · Reviewer_gsmP · 2021-11-25
> > > > > **Thanks for the clarification**
> > > > >
> > > > > For me, the [*] part is still not very clear. Their \gamma_I seems to have 0 support somewhere but since your y_i is an ergodic distribution it should have full support. Anyway, it might be not easy to describe things clearly here and I buy the divine intervention and am ok if you want to defer the burden of the proof to Santosh Vempala (2005) during the review process. I'll increase my score accordingly but I do hope this technique is clearly and explicitly documented in the next version such that it is no longer a folklore fact for the RL community. I can image this technique can be used in analyzing many RL algorithms and if you have it documented clearly, I think this work is likely to get more citation.

---

> > > > > > ### Author Response · Authors · 2021-11-27
> > > > > > **Thank you; we feel we have a complete proof now.**
> > > > > >
> > > > > > Dear gsmP,
> > > > > >
> > > > > > We thank you for your further detailed feedback, which continues to greatly improve our work;
> > > > > > we also thank you for your kind support.
> > > > > >
> > > > > > We agree with you that our present situation with the invocation of "divine
> > > > > > intervention" is unsatisfactory.  Therefore we set out to produce a complete proof;
> > > > > > instead, we found a way to directly apply the "optimal coupling theorem" (rewriting the total variation
> > > > > > in terms of $\Pr[X\neq Y]$ for a coupled pair $(X,Y)$ that minimizes the corresponding
> > > > > > expression), and can completely omit the earlier complicated reasoning which defined and introduced
> > > > > > $(E_j,y_j,z_j)$.
> > > > > >
> > > > > > We give the proof below in detail, but here is a sketch of the main ideas.
> > > > > > Consider the step of the proof just before we use $E_j$ to replace
> > > > > > the stochastic Bellman operator, $\mathcal T_j$ , with the expected Bellman operator
> > > > > > at stationarity, $\mathcal T_{\mathfrak{s}}$.
> > > > > > Instead of using $E_j$, we can simply add and subtract $\mathcal T_{\mathfrak{s}}$.
> > > > > > The key is that we now have a difference $\mathcal T_j - \mathcal T_{\mathfrak{s}}$,
> > > > > > and we can use optimal coupling to couple the randomness
> > > > > > in these two operators and argue that their difference is small as
> > > > > > a consequence of the small total variation of their randomness.
> > > > > >
> > > > > > We thank you once again for all your detailed comments, support, and patience.
> > > > > > You suggested that it would be valuable to not only produce such a proof, but to make it
> > > > > > easy to use in other works; we will work to split out such convenient lemmas from our
> > > > > > current lengthy TD analysis.
> > > > > >
> > > > > >
> > > > > > ---
> > > > > >
> > > > > >
> > > > > > As mentioned, consider the bottom equation block of page 24 in our submission.
> > > > > > Starting with the original left hand side, write
> > > > > > $$
> > > > > >   \mathbb{E}\langle u_{j-k} - \bar u, \mathcal T_j(u_{j-k})\rangle
> > > > > >   =
> > > > > >   \mathbb{E}\langle u_{j-k} - \bar u, \mathcal T_{\mathfrak{s}}(u_{j-k})\rangle
> > > > > >   +
> > > > > >   \mathbb{E}\langle u_{j-k} - \bar u, \mathcal T_j(u_{j-k}) - \mathcal T_{\mathfrak{s}}(u_{j-k})\rangle,
> > > > > > $$
> > > > > > and henceforth focus only on bounding the second term, since the first term is exactly the expression
> > > > > > we use in the rest of the proof of Lemma C.4.
> > > > > >
> > > > > > Let $\mathbb E_{j-k}$ denote a conditional expectation which conditions on all randomness in $u_{j-k}$,
> > > > > > and let $(x,x',r)$ denote the corresponding random variables in $\mathcal T_j$.  Similarly, let $\mathbb E_{\mathfrak{s}}$
> > > > > > denote the expectation over the randomness in $\mathcal T_{\mathfrak{s}}$, and let $(y,y',r')$ denote
> > > > > > the corresponding random variables.  By the tower property of conditional expectation and the
> > > > > > definition of these two Bellman operators in terms of the preceding notation,
> > > > > > $$
> > > > > > \begin{aligned}
> > > > > >   \mathbb{E}&\langle u_{j-k} - \bar u, \mathcal T_j(u_{j-k}) - \mathcal T_{\mathfrak{s}}(u_{j-k})\rangle
> > > > > >   \\\\
> > > > > >   =&
> > > > > >   \mathbb{E}\mathbb E_{j-k}\langle u_{j-k} - \bar u, (x x^T - \gamma x (x')^T )u_{j-k} - xr - \mathbb E_{\mathfrak{s}} [(yy^T - \gamma y (y')^T)u_{j-k}  - yr']\rangle
> > > > > >   \\\\
> > > > > >   =&
> > > > > >   \mathbb{E}\left\langle u_{j-k} - \bar u, ( \mathbb E_{j-k} x x^T - \gamma x (x')^T )u_{j-k} - ( \mathbb E_{\mathfrak{s}} yy^T - \gamma y (y')^T) u_{j-k}\right\rangle
> > > > > >   \\\\
> > > > > >   &\qquad+
> > > > > >   \mathbb{E}\left\langle u_{j-k} - \bar u, \mathbb E_{j-k} xr  - \mathbb E_{\mathfrak{s}} yr'\right\rangle.
> > > > > > \end{aligned}
> > > > > > $$
> > > > > > Henceforth we will control only the first term; the second term, which concerns the rewards, follows
> > > > > > via similar but easier reasoning.
> > > > > >
> > > > > > Now comes the key step.  Let $\pi$ denote an optimal coupling of the random variables $(x,x',r)$ conditioned on $u_{j-k}$,
> > > > > > coupled with $(y,y',r')$ at stationarity.
> > > > > > Since $\mathbb E_{j-k}$ and $\mathbb E_{\mathfrak{s}}$ appear in two separate non-interacting expressions above,
> > > > > > and since the randomness in those two distributions matches the corresponding marginals of $\pi$,
> > > > > > we can replace these two expectations with $\mathbb E_{\pi}$.
> > > > > > Continuing, and additionally using the Cauchy-Schwarz inequality, and letting $\\|\\cdot\\|\_{\\text{op}}$ denote
> > > > > > the spectral norm of a matrix,
> > > > > > $$
> > > > > > \begin{aligned}
> > > > > >   &
> > > > > >   \mathbb{E}\left\langle u_{j-k} - \bar u, ( \mathbb E_{j-k} x x^T - \gamma x (x')^T )u_{j-k} - ( \mathbb E_{\mathfrak{s}} yy^T - \gamma y (y')^T) u_{j-k}\right\rangle
> > > > > >   \\\\
> > > > > >   =&
> > > > > >   \mathbb{E}\left\langle u_{j-k} - \bar u, \left( ( \mathbb E_{\pi} x x^T - \gamma x (x')^T ) - ( \mathbb E_{\pi} yy^T - \gamma y (y')^T)\right) u_{j-k}\right\rangle
> > > > > >   \\\\
> > > > > >   \geq&
> > > > > >   -\mathbb{E}\\| u_{j-k} - \bar u\\|\cdot \\|u_{j-k}\\| \cdot \left\\| \mathbb E_{\pi} x x^T - \gamma x (x')^T - yy^T + \gamma y (y')^T \right\\|\_{\\text{op}}.
> > > > > > \end{aligned}
> > > > > > $$
> > > > > > To control the first two terms, we can use reasoning in the existing proof.
> > > > > > For the last term,
> > > > > > we use the optimal coupling property: with probability $1 - \text{TV}((x,x',r), (y,y',r'))$, all terms cancel
> > > > > > and we get exactly zero; otherwise, with probability $\text{TV}((x,x',r), (y,y',r'))$ (which is small),
> > > > > > we control the expression in a worst-case way (as in the existing proof).
> > > > > >
> > > > > > This will change some constants in the proof, but all other steps can proceed as before.

---

> > > > > > > ### Comment · Reviewer_gsmP · 2021-11-29
> > > > > > > **Thanks for the clarification**
> > > > > > >
> > > > > > > Thanks for providing the details. I think it now makes sense to me.

---

> > > > > > > > ### Author Response · Authors · 2021-11-30
> > > > > > > > **Thank you!**
> > > > > > > >
> > > > > > > > Thank you for all of the time you have committed, and for your thoughtful discussion and support throughout; we are indebted to you for helping us create a much stronger submission!

---

> > > > > > > > ### Public Comment · ~Matus_Telgarsky1 · 2022-03-13
> > > > > > > > **Thank you**
> > > > > > > >
> > > > > > > > Thank you once again for your detailed reading of our paper and the many useful comments.
> > > > > > > >
> > > > > > > > In the camera ready, I followed the fix laid out above, using the coupling characterization of total variation directly.
> > > > > > > >
> > > > > > > > That said, I was still unsatisfied with the proof, partly because I had originally intended to make a high-probability guarantee.  I found a different proof structure which can provide such a guarantee, and also handle SGD with the same proof scheme.  If this sounds interesting to you, you can find it at https://arxiv.org/abs/2202.06915 .  I really appreciate your comments and hope we can chat further some day.

---

> > > ### Comment · Reviewer_gsmP · 2021-11-24
> > > **Previous work requires starting TD from the stationary distribution of the policy**
> > >
> > > I tend to disagree with this argument.
> > > I understand that Bhandari et al, 2018 have this assumption in their Assumption 1. But [x] does not have this assumption. [x] extend Bhandari et al, 2018 from linear TD for policy evaluation to linear SARSA for control -- linear TD is a special case for linear SARSA by setting the policy improvement operator to output a constant policy. So I believe [x] is a superset of Bhandari et al, 2018. I'm quite familiar with [x] and do not think [x] uses this assumption.
> > >
> > > [x] Zou, Shaofeng, Tengyu Xu, and Yingbin Liang. "Finite-sample analysis for sarsa with linear function approximation." arXiv preprint arXiv:1902.02234 (2019).

---

> > > > ### Author Response · Authors · 2021-11-25
> > > > **Thank you for the new reference!**
> > > >
> > > > Dear reviewer gsmP,
> > > >
> > > > Thank you for this very interesting reference "[x]"!  We had unfortunately
> > > > overlooked it before, but certainly agree that it does not assume stationarity.
> > > > As such, we should revert the emphasis we added in the revision, which stated
> > > > that not requiring stationarity is one of our contributions; instead, we should
> > > > present our TD analysis's contributions as in the original submission, namely
> > > > highlighting the lack of projections, but also include a citation to [x], and
> > > > discuss it and its assumptions relative to Bhandari et al in detail.
> > > >
> > > > There is one technical point to discuss when referring to Bhandari et al and to
> > > > [x]. As far as we can see, all the results in [x] are under [x]'s "Assumption
> > > > 2", an eigenvalue condition which in pure optimization terms is analogous to
> > > > strong convexity/concavity, and ensures the updates are contractive, and leads
> > > > to a 1/t rate.  Bhandari et al have a similar eigenvalue condition (their "$\omega$",
> > > > which is assumed positive on page 6 of their paper), though in their finite-time
> > > > guarantee (Theorem 4), while $\omega$ appears in (b) and (c) (where (c) also gets
> > > > a $1/t$ rate as in [x]), it does not appear in (a) and it is not clear it is used
> > > > (we have not checked the proofs).  This eigenvalue condition can be false in our setting
> > > > (basically, since the maximum entropy optimal policy can place zero probability on some actions),
> > > > so we made no such assumption, and thus our TD guarantee looks more like (a) of Bhandari et al's
> > > > Theorem 4.  Overall our plan when revising to include [x]
> > > > is to
> > > > treat this eigenvalue condition as a technical point and otherwise not emphasize it,
> > > > but if the reviewer feels we should highlight it in some way, we are eager to discuss this
> > > > point further and adjust our submission appropriately!
> > > >
> > > > Thank you once again for your precise comments and for the new reference [x];
> > > > these changes will greatly improve our submission!

---

> > > > > ### Comment · Reviewer_gsmP · 2021-11-25
> > > > > **Thanks for the clarification**
> > > > >
> > > > > I agree that the Assumption 2 of [x] can be falsified when some action has 0 probability. It is great to see this work does not need to make such an assumption.

---

> > > > > > ### Author Response · Authors · 2021-11-27
> > > > > > **Thank you!**
> > > > > >
> > > > > > Dear gsmP,
> > > > > >
> > > > > > Thank you for your detailed technical comments and support.  We will aim to be precise when discussing these assumptions in our revisions.  We're currently at the page limit (and had to delete some of our old open problems!), but will find a way.  Thank you once again!

---

### Official Review · Reviewer_3pPx · 2021-10-28

**Correctness:** 4
**Technical Novelty And Significance:** 4
**Empirical Novelty And Significance:** Not applicable
**Recommendation:** 8
**Confidence:** 2

**Main Review:**

With a very slow convergence rate 1/epsilon^{14}, the provided algorithm is strictly theoretical and is not intended to be applied (the authors are clear and honest on that ground).
Despite the real author's efforts, the paper is very hard to read and requires an advanced mathematical background coupled with a vast culture on theoretical aspects of RL. The main paper and the few pages of proof that I checked seemed solid but I must admit that this paper is behind my reach.
That said, I find interesting for the community to get a few theoretical results on actor-critic, and I do not expect such results to be easy to handle.

Minor remarks:
page 12 line 3 "is just ... and need not be" -> is just ... and needs not to be"


**Summary Of The Paper:**

This heavily technical theoretical paper main contribution is to show that Natural Actor-Critic is "philosophically" implicitly biased toward high entropy policies.
More precisely, the key result is to show that Actor-Critic, when set as a batched  mirror descent algorithm applied on a linear finite states and linear finite actions MDP with linear softmax policies, without regularization or explicit exploration, but with properly chosen parameters, maintains its policies in a KL ball of radius 1+ ln(k) + 1/(1-gamma)^2 around the maximum entropy optimal policy (the optimal policy which samples uniformly the optimal actions) with high probability (Theorem 1.4).
It is worth noting that this result is obtained without global mixing assumptions on the MDP (only with a mixing assumption on the target policy).
This is an improvement from previous work (Khodadadian 2021) who assumed uniform mixing and provided only expectation bounds.

Dropping these assumption is however costly: with a step size set to 1/sqrt(t), the number of iterations required to obtain and epsilon-optimal policy is epsilon^{-14}.

The core of the paper is in the detailed proof which spans 11 pages of appendix.

**Summary Of The Review:**

A highly theoretical paper underlying an interesting result on the bias of actor critic toward high entropy policies. The proofs are long and technical and assessing their correctness is behind my reach. But my bet is that it is a good and correct paper.

---

> ### Author Response · Authors · 2021-11-18
> **Thank you for your review.**
>
> We thank the reviewer for their comments, time, and support.  As follows are our main comments, and
> the corresponding changes in our revised submission (where applicable):
>
> - **"Despite the real author's efforts, the paper is very hard to read and requires an advanced mathematical background coupled with a vast culture on theoretical aspects of RL".** We have made extension revisions and tried hard to improve the presentation of the paper, as partially discussed
> in our general response to all reviewers above, but as verified more easily via openreview's diff feature.  If the reviewer has any further comments on presentation, we will gladly work towards addressing them.
>
> - **"Dropping these assumption is however costly: [...] the number of iterations required to obtain and $\epsilon$-optimal policy is $\epsilon^{-14}$".**
>   We agree this is slow, and in particular too slow for practice and slower than many related works.
>   However, in addition to our goal of establishing that the standard assumptions are not necessary,
>   we also feel there is great promise for faster rates.  For example, we presented two mirror
>   descent bounds, and could only use one due to statistical dependency issues; if some future work
>   could overcome these issues, a faster rate would come immediately.  As another example,
>   our TD analysis (and others in the literature) lack proper high probability bounds;
>   fixing this would again improve the rate.
>   Therefore we feel there is a clear path for future
>   work to build upon ours, and further improve the rate.  If the reviewer finds these remarks
>   interesting, we can better document the most promising places for improvement in our revisions,
>   though at present our revisions have already caused us to reach the full expanded 9-page limit!
>
>   (Briefly, we remark that in the process of removing mini-batching from TD and making various
>   other revisions, our rate has worsened from $1/\epsilon^{14}$ to $1/\epsilon^{16}$.)
>
> - **"page 12 line 3 [...]".** Thank you, we have fixed this.
>
> We thank the reviewer once again, and look forward to further comments, and to the resulting improvements to our submission!

---

### Official Review · Reviewer_UirM · 2021-11-02

**Correctness:** 3
**Technical Novelty And Significance:** 3
**Empirical Novelty And Significance:** Not applicable
**Recommendation:** 6
**Confidence:** 3

**Main Review:**

The paper is well written, fairly simple to follow, and provides good justifications and reminders about the overall story to the reader when going into mathematical details. I was able to follow the main story but not the specific details for proving the different lemmas. Overall, I think the motivation is strong and a cleaner analysis is definitely much appreciated. On the downside, there aren't any algorithmic insights I could gather from the paper. Particularly, is there anything a practitioner can take away from this? This might not be a concern if the paper was offering a novel result, however the question in this case is what exactly do we gain from this alternate analysis. Right now, I don't think we learn much but hopefully the authors can comment more on this point. Finally, below are some questions I hope the authors can reply to for me to understand the work better:

- So the assumption that the optimal maximum entropy policy places a positive mass on every state + the KL bound allows us to use the same assumption on any policy within the KL bound. How should one contrast this with the assumption that the softmax parameterization itself places a positive mass on each state (+ is that even true)?

- The KL is upper bounded by the number of actions k. Can you say something about how growing number of actions affects the KL bound? Moreover, wouldn't the horizon term dominate for common cases here? How tight is this KL bound overall? Intuitively, it seems like the KL ball should be growing smaller as we explore more of the MDP, whereas here it is fixed across all 't'. Can you provide an intuitive explanation for why this is still fine?

**Summary Of The Paper:**

The paper provides a novel analysis to the linear MDP setting through an actor critic setup where the policy is softmax-parameterized. The claim is that we can avoid mixing time and exploration based assumptions by using this analysis. This is attractive since its much simpler/cleaner, but also comes with a slower convergence rate. The main steps include bounding the policy within a KL ball of the maximum entropy optimal policy through a mirror descent style analysis, followed by controlling the critic approximation error given the policy is within the KL ball.

**Summary Of The Review:**

The paper is nicely written and provides an interesting alternative to analysing a linear MDP with an actor critic setup. I have some doubts about how much insight can be gained from this analysis for future work, especially algorithmically and therefore I am currently advocating for a weak accept.

---

> ### Author Response · Authors · 2021-11-18
> **Thank you for your review.**
>
> We thank the reviewer for their comments, time, and support.  As follows are our main comments, and
> the corresponding changes in our revised submission (where applicable):
>
> - **"I was able to follow the main story but not the specific details for proving the different lemmas".** We have ensured that the main ideas of our proofs are now present in the body of the paper, and also expanded proofs in the appendix, and would gladly hear further comments on this point!
>
> - **"Particularly, is there anything a practitioner can take away from this?"**  We agree
>   with the reviewer that there is no immediate benefit to practitioners, and also agree that this
>   is a valuable eventual goal.  For now, we feel that (a) our analysis is a first step towards
>   analysis and design of algorithms which are closer to ones used in practice, since, unlike
>   prior work, we remove many subroutines and assumptions,
>   and (b) we provide a sanity check that a simple algorithm can suffice in the setting we study,
>   and (c) the theoretical literature often suggests explicit exploration is necessary (which would
>   have practical consequences), whereas our work suggests it is not.
>
> - **"The optimal maximum entropy policy places a positive mass on every state [...]  How should one contrast this with the assumption that the softmax parameterization itself places a positive mass on each state".**  It is true that a softmax policy in this setting will always place positive
>   mass on every state, and moreover on every action in every state.  However,
>   in general there is no lower bound on the magnitude of this positive probability mass.
>   Consequently, the KL between the maximum entropy optimal policy
>   and arbitrary policies is unbounded, and in stark contrast with our guarantee that the algorithm
>   only considers policies with bounded KL despite using no regularization or constraints.
>   In our revisions, we have included comments on this in Remark 1.5 after the main theorem.
>
> - Regarding **"the KL is upper bounded by the number of actions k"**, and other comments raised
>   in the reviewer's second bullet.  As a quick clarification, we assume you mean that our bound
>   has a term $\ln k$, which depends on the number of actions $k$, and that the other
>   term $1/ (1-\gamma)^2$ is a "horizon term".  Firstly, we agree that $\ln k$ is problematic
>   if the action space is infinite; however, since the dependence is only logarithmic, we hope
>   that the our proof scheme can be later extended to handle infinite action spaces under
>   reasonable assumptions.
>   Secondly, we agree that since $\gamma$ is typically close to $1$, the
>   "horizon term" will dominate; it would also be nice in future work to handle this term,
>   though we feel that this will require new ideas on top of our present analysis.
>   Lastly, regarding "the KL ball should be growing smaller as we explore more of the MDP",
>   does the
>   reviewer mean that over time we should converge to the maximum entropy optimal policy in KL
>   divergence, or do they refer to some other measure of uncertainty?
>   Our bound holds for all time and thus
>   in particular pays for early iterations where knowledge of the MDP is bad.  However, our proof
>   and assumptions are not strong enough to give convergence to the maximum entropy optimal policy,
>   though we agree this is interesting.  We will gladly revise our submission further after
>   further comments from the reviewer.
>
> We thank the reviewer once again, and look forward to further comments, and to the resulting improvements to our submission!

---

> > ### Comment · Reviewer_UirM · 2021-11-30
> > **Thank you for the response!**
> >
> > Regarding the point on the size of the KL ball, I was referring to convergence to the maximum entropy optimal policy. Thanks for clarifying this. Overall, I am happy to see this paper accepted.

---

> > > ### Author Response · Authors · 2021-12-03
> > > **Thank you!**
> > >
> > > Thank you for taking another look, and for clarifying!  This particular point about convergence to a specific policy (and not just the high entropy set of policies) is very interesting to us, and in our revisions we will try to recoup space so that we can discuss it in the open problems.  Based on both preliminary experiments and preliminary proofs, we believe this convergence is in fact false in general, so it is interesting to wonder if the algorithm can be modified to ensure this, and whether such a modification is even worthwhile more generally (e.g., practical consequences).
> > >
> > > Thank you once again for your comments and support!

---

### Author Response · Authors · 2021-11-18
**Revision summary.**

We thank the reviewers for their comments, time, and support.  We have heavily revised
the paper to address reviewer comments, to improve clarity, and to improve various
technical aspects.

Our main changes are summarized as follows.

- We have removed the use of mini-batches in the TD step, both to improve readability
  (our original algorithm needed triple subscripts),
  and to allow direct comparison to prior TD work.
  Regarding this comparison to prior TD work,
  we have completed a more careful literature survey,
  and can say that prior work not only requires projections as we discussed originally,
  but additionally it requires starting TD from the stationary distribution of the policy,
  whereas we allow it to be run from an arbitrary state.

  We note additionally that reviewer gsmP had correctness concerns in our original TD proof.
  Despite dropping mini-batches,
  all the ideas in the new proof are the same as the old one;
  the new proof is twice as long, but that is because we tried hard to make
  everything explicit and to make the proof well-organized and readable.

- In response to comments by reviewers gsmP and UirM, we have included more proof
  sketches in the body of the paper, allowing at least a rough correctness
  check to be performed without looking at the appendices.  We agree this was
  a bad omission on our part and hope the reviewers like the changes.

We thank the reviewers for their careful comments,
and look forward to further discussion!

---

> ### Author Response · Authors · 2021-11-23
> **(Minor revision comment.)**
>
> We cleaned a few lines of algebra in the TD proof but performed no other changes; as such, we went ahead and deleted the November 17 revision, since we felt the important comparison is against the original submission, while keeping the November 17 revision ran a risk of confusion.
>
> We apologize for the extra notifications these revisions have caused; we look forward to further discussion!

---

> > ### Public Comment · ~Matus_Telgarsky1 · 2022-03-13
> > **(Camera ready revision comment.)**
> >
> > Between the feedback phase and camera ready, while we did a full polishing pass of the body of the paper, our focus was on improving the proofs in the appendices.  As identified by reviewer gsmP, there was a sequence of unclear steps in the TD analysis; we have now fixed this oversight using the plan we laid out in the discussion phase comments.   We are greatly indebted to reviewer gsmP and the other ICLR reviewers for greatly improving our work!

---

### Decision · Program_Chairs · 2022-01-20

**Decision:**

Accept (Poster)

**Comment:**

The paper makes a significant contribution in the rather sparse and challenging field of convergence analyses of actor-critic style algorithms, under the linear MDP structural assumption, showing that there is a natural bias towards being high-entropy. As one of the reviewers points out, although it is unlikely that the strategy actually proposed is amenable to implementation, the paper nevertheless provides a clean and novel analysis of convergence of learning by eschewing the usual mixing time type assumptions often found in the theoretically-oriented RL literature. Based on this strength of the paper, I am glad to recommend its acceptance.